# Universal Prompt Tuning for Graph Neural Networks

**Taoran Fang**[1], **Yunchao Zhang**[1], **Yang Yang**[1]*, **Chunping Wang**[2], **Lei Chen**[2]
[1]Zhejiang University, [2]FinVolution Group
{fangtr,3190105622,yangya}@zju.edu.cn,
{wangchunping02,chenlei04}@xinye.com

## Abstract

In recent years, prompt tuning has sparked a research surge in adapting pre-trained models. Unlike the unified pre-training strategy employed in the language field, the graph field exhibits diverse pre-training strategies, posing challenges in designing appropriate prompt-based tuning methods for graph neural networks. While some pioneering work has devised specialized prompting functions for models that employ edge prediction as their pre-training tasks, these methods are limited to specific pre-trained GNN models and lack broader applicability. In this paper, we introduce a universal prompt-based tuning method called *Graph Prompt Feature (GPF)* for pre-trained GNN models under any pre-training strategy. GPF operates on the input graph's feature space and can theoretically achieve an equivalent effect to any form of prompting function. Consequently, we no longer need to illustrate the prompting function corresponding to each pre-training strategy explicitly. Instead, we employ GPF to obtain the prompted graph for the downstream task in an adaptive manner. We provide rigorous derivations to demonstrate the universality of GPF and make guarantee of its effectiveness. The experimental results under various pre-training strategies indicate that our method performs better than fine-tuning, with an average improvement of about $1.4\%$ in full-shot scenarios and about $3.2\%$ in few-shot scenarios. Moreover, our method significantly outperforms existing specialized prompt-based tuning methods when applied to models utilizing the pre-training strategy they specialize in. These numerous advantages position our method as a compelling alternative to fine-tuning for downstream adaptations. Our code is available at: `https://github.com/zjunet/GPF`.

## 1   Introduction

Graph neural networks (GNNs) have garnered significant attention from researchers due to their remarkable success in graph representation learning [Kipf and Welling, 2017, Hamilton et al., 2017, Xu et al., 2019]. However, two fundamental challenges hinder the large-scale practical applications of GNNs. One is the scarcity of labeled data in the real world [Zitnik et al., 2018], and the other is the low out-of-distribution generalization ability of the trained models [Hu et al., 2020a, Knyazev et al., 2019, Yehudai et al., 2021, Morris et al., 2019]. To overcome these challenges, researchers have made substantial efforts in designing pre-trained GNN models [Xia et al., 2022b, Hu et al., 2020a,b, Lu et al., 2021] in recent years. Similar to the pre-trained models in the language field, pre-trained GNN models undergo training on extensive pre-training datasets and are subsequently adapted to downstream tasks. Most existing pre-trained GNN models obey the "pre-train, fine-tune" learning strategy [Xu et al., 2021a]. Specifically, we train a GNN model with a massive corpus of pre-training graphs, then we utilize the pre-trained GNN model as initialization and fine-tune the model parameters based on the specific downstream task.

---

*Corresponding author.

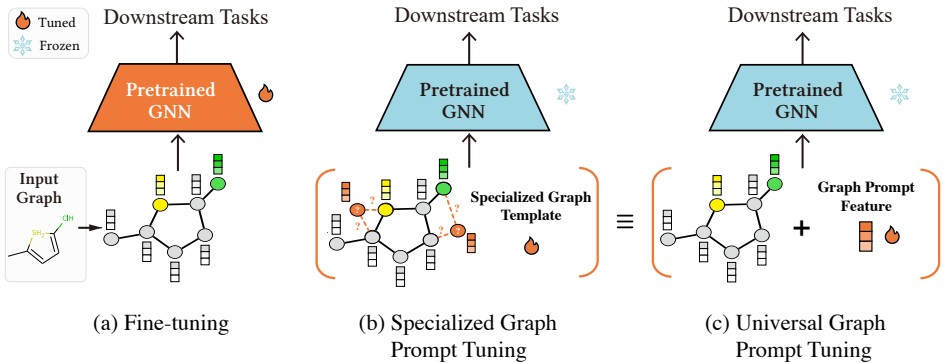

(a) Fine-tuning    (b) Specialized Graph Prompt Tuning    (c) Universal Graph Prompt Tuning

Figure 1: **Comparison of universal graph prompt tuning and existing approaches.** (a) Fine-tuning updates the parameters of the pre-trained GNN model. (b) Existing specialized prompt-based tuning methods generate manual graph templates to adapt the models under certain pre-training strategies. (c) Our universal graph prompt tuning works on the feature space of the input graph. It can achieve an equivalent effect to any form of prompting function and be applied to any pre-trained GNN model.

However, the "pre-train, fine-tune" framework of pre-trained GNN models also presents several critical issues [Jin et al., 2020]. First, there is a misalignment between the objectives of pre-training tasks and downstream tasks [Liu et al., 2022a]. Most existing pre-trained models employ self-supervised tasks [Liu et al., 2021e] such as edge prediction and attribute masking as the training targets during pre-training, while the downstream tasks involve graph or node classification. This disparity in objectives leads to sub-optimal performance [Liu et al., 2023]. Additionally, ensuring that the model retains its generalization ability is challenging. Pre-trained models may suffer from catastrophic forgetting [Zhou and Cao, 2021, Liu et al., 2020] during downstream adaptation. This issue becomes particularly acute when the downstream data is small in scale, approaching the few-shot scenarios [Zhang et al., 2022]. The pre-trained model tends to over-fit the downstream data in such cases, rendering the pre-training process ineffective.

*"If your question isn't getting the desired response, try rephrasing it."* In recent years, a novel approach called prompt tuning has emerged as a powerful method for downstream adaptation, addressing the aforementioned challenges. This technique has achieved significant success in Natural Language Processing [Li and Liang, 2021, Lester et al., 2021a, Liu et al., 2022a,b] and Computer Vision [Bahng et al., 2022, Jia et al., 2022]. Prompt tuning provides an alternative method for adapting pre-trained models to specific downstream tasks: it freezes the parameters of the pre-trained model and modifies the input data. Unlike fine-tuning, prompt tuning diverges from tuning the parameters of the pre-trained model and instead focuses on adapting the data space by transforming the input.

Despite that, applying prompt tuning on pre-trained GNN models poses significant challenges and is far from straightforward. First, the diverse pre-training strategies employed on graphs make it difficult to design suitable prompting functions. Previous research [Liu et al., 2022a] suggests that the prompting function should be closely aligned with the pre-training strategy. For pre-trained language models, the typical pre-training tasks involve masked sentence completion [Brown et al., 2020]. In order to align with this task, we may modify a sentence like "I received a gift" to "I received a gift, and I feel [Mask]" to make it closer to the task of sentence completion. However, in the case of graph pre-training, there is no unified pre-training task, making it challenging to design feasible prompting functions. Some pioneering studies [Sun et al., 2022, Liu et al., 2023] have applied prompt-based tuning methods to models pre-trained by edge prediction [Kipf and Welling, 2016b]. They introduce virtual class-prototype nodes/graphs with learnable links into the original graph, making the adaptation process more akin to edge prediction. However, these methods have limited applicability and are only compatible with specific models. When it comes to more intricate pre-training strategies, it becomes challenging to design manual prompting functions in the same manner as employed for link prediction. Consequently, no prompt-based tuning method is available for models pre-trained using alternative strategies, such as attribute masking [Hu et al., 2020a]. Furthermore, existing prompt-based tuning methods for GNN models are predominantly designed based on intuition, lacking theoretical guarantees for their effectiveness.

In this paper, we address the aforementioned issues for graph prompt tuning. To deal with the diversity of graph pre-training strategies, we propose a universal prompt-based tuning method that can be

applied to the pre-trained GNN models that employ any pre-training strategy. Figure 1 illustrates the distinction between our universal prompt-based tuning method and existing approaches. Our solution, called Graph Prompt Feature (GPF), operates on the input graph's feature space and involves adding a shared learnable vector to all node features in the graph. This approach is easily applicable to any GNN architecture. We rigorously demonstrate that GPF can achieve comparable results to any form of prompting function when applied to arbitrary pre-trained GNN models. Consequently, instead of explicitly illustrating the prompting function corresponding to each pre-training strategy, we adopt GPF to dynamically obtain the prompted graph for downstream tasks. We also introduce a theoretically stronger variant of GPF, named GPF-plus, for practical application, which incorporates different prompted features for different nodes in the graph. To guarantee the effectiveness of our proposed GPF and GPF-plus, we provide theoretical analyses to prove that GPF and GPF-plus are not weaker than full fine-tuning and can obtain better theoretical tuning results in some cases. Furthermore, we conduct extensive experiments to validate the efficacy of our methods. Despite using a significantly smaller number of tunable parameters than fine-tuning, GPF and GPF-plus achieve better results across all pre-training strategies. For models pre-trained using edge prediction, GPF and GPF-plus exhibit a substantial performance advantage over existing specialized prompt-based tuning methods. Overall, the contributions of our work can be summarized as follows:

- To the best of our knowledge, we present the first investigation of universal prompt-based tuning methods for existing pre-trained GNN models. We propose GPF and its variant, GPF-plus, as novel approaches for universal graph prompt tuning. Our methods can be applied to the pre-trained GNN models that employ any pre-training strategy.

- We provide theoretical guarantees for the effectiveness of GPF and GPF-plus. We demonstrate that GPF and GPF-plus can achieve an equivalent effect to any prompting function and can obtain better tuning results in some cases compared to fine-tuning.

- We conduct extensive experiments (both full-shot and few-shot scenarios) to validate the effectiveness of GPF and GPF-plus. The experimental results indicate that GPF and GPF-plus can perform better than fine-tuning, with an average improvement of about $1.4\%$ in full-shot scenarios and about $3.2\%$ in few-shot scenarios. Furthermore, GPF and GPF-plus significantly outperform existing prompt-based tuning methods when applied to models that utilize the pre-training strategy they specialize in.

## 2 Related work

**Pre-trained GNN Models**  Inspired by the remarkable achievements of pre-trained models in Natural Language Processing [Qiu et al., 2020b] and Computer Vision [Long et al., 2022], substantial efforts have been dedicated to pre-trained GNN models (PGMs) [Xia et al., 2022b] in recent years. These methods utilize self-supervised strategies [Jin et al., 2020] to acquire meaningful representations from extensive pre-training graphs. GAE [Kipf and Welling, 2016a] first uses edge prediction as the objective task to train graph representations. Deep Graph Infomax (DGI) [Velickovic et al., 2019b] and InfoGraph [Sun et al., 2019] are proposed to garner nodes or graph representations by maximizing the mutual information between graph-level and substructure-level representations of different granularity. Hu et al. [2020a] employ attribute masking and context prediction as pre-training tasks to predict molecular properties and protein functions. Both GROVER [Rong et al., 2020] and MGSSL [Zhang et al., 2021a] propose to predict the presence of the motifs or generate them with the consideration that rich domain knowledge of molecules hides in the motifs. Graph Contrastive Learning (GCL) is another widely adopted pre-training strategy for GNN models. GraphCL [You et al., 2020] and JOAO [You et al., 2021] propose various augmentation strategies to generate different augmented views for contrastive learning. In summary, there exists a diverse range of pre-training strategies for GNN models, each characterized by unique objectives.

**Prompt-based Tuning Methods**  Prompt-based tuning methods, originating from Natural Language Processing, have been widely used to facilitate the adaptation of pre-trained language models to various downstream tasks [Liu et al., 2021a]. Research has also explored the design of soft prompts to achieve optimal performance [Lester et al., 2021b, Liu et al., 2021c]. These methods freeze the parameters of the pre-train models and introduce additional learnable components in the input space, thereby enhancing the compatibility between inputs and pre-trained models. Aside from the success of prompts in the language field, the prompting methods are utilized in other areas. Jia et al. [2022] and Bahng et al. [2022] investigate the efficacy of adapting large-scale models in the vision field by

modifying input images at the pixel level. In the realm of graph neural networks, the exploration of prompt-based tuning methods is still limited. Some pioneering work [Sun et al., 2022, Liu et al., 2023] applies prompt-based tuning methods on the models pre-trained by edge prediction [Kipf and Welling, 2016b]. These methods introduce virtual class-prototype nodes/graphs with learnable links into the input graph, making the downstream adaptation more closely resemble edge prediction. However, these methods are specialized for models pre-trained using edge prediction and cannot be applied to models trained with other strategies. We are the first to investigate the universal prompt-based tuning methods that can be applied to the GNN models under any pre-training strategy.

## 3 Methodology

We introduce *graph prompt tuning* for adapting pre-trained GNN models to downstream tasks. It is important to note that there are several types of downstream tasks in graph analysis, including node classification, link prediction, and graph classification. We first concentrate on the graph classification task and then extend our method to node-wise tasks. We define the notations in Section 3.1, then illustrate the process of graph prompt tuning in Section 3.2. We introduce our universal graph prompt tuning method in Section 3.3 and provide theoretical analyses in Section 3.4. Finally, we present the extension of our method to node-wise tasks (node classification and link prediction) in the appendix.

### 3.1 Preliminaries

Let $\mathcal{G} = (\mathcal{V}, \mathcal{E}) \in \mathbb{G}$ represents a graph, where $\mathcal{V} = \{v_1, v_2, \ldots, v_N\}$, $\mathcal{E} \subseteq \mathcal{V} \times \mathcal{V}$ denote the node set and edge set respectively. The node features can be denoted as a matrix $\mathbf{X} = \{x_1, x_2, \ldots, x_N\} \in \mathbb{R}^{N \times F}$, where $x_i \in \mathbb{R}^F$ is the feature of the node $v_i$, and $F$ is the dimensionality of node features. $\mathbf{A} \in \{0, 1\}^{N \times N}$ denotes the adjacency matrix, where $\mathbf{A}_{ij} = 1$ if $(v_i, v_j) \in \mathcal{E}$.

**Fine-Tuning Pre-trained Models.** Given a pre-trained GNN model $f$, a learnable projection head $\theta$ and a downstream task dataset $\mathcal{D} = \{(\mathcal{G}_1, y_1), \ldots, (\mathcal{G}_m, y_m)\}$, we adjust the parameters of the pre-trained model $f$ and the projection head $\theta$ to maximize the likelihood of predicting the correct labels $y$ of the downstream graph $\mathcal{G}$:

$$\max_{f,\theta} P_{f,\theta}(y|\mathcal{G}) \tag{1}$$

### 3.2 Graph Prompt Tuning

**Overall Process.** Our proposed *graph prompt tuning* works on the input space by drawing on the design of the prompt tuning in the language field [Liu et al., 2022a]. Given a frozen pre-trained GNN model $f$, a learnable projection head $\theta$, and a downstream task dataset $\mathcal{D} = \{(\mathcal{G}_1, y_1), \ldots, (\mathcal{G}_m, y_m)\}$, our target is to obtain a task-specific *graph prompt* $g_\phi \colon \mathbb{G} \to \mathbb{G}$ parameterized by $\phi$. The *graph prompt* $g_\phi(\cdot)$ transforms the input graph $\mathcal{G}$ into a specific *prompted graph* $g_\phi(\mathcal{G})$. And then $g_\phi(\mathcal{G})$ will replace $\mathcal{G}$ as input to the pre-trained GNN model $f$. During the downstream task training, we select the optimal parameters of $\phi$ and $\theta$ that maximize the likelihood of predicting the correct labels $y$ without tuning the pre-trained model $f$, which can be formulated as:

$$\max_{\phi,\theta} P_{f,\theta}(y|g_\phi(\mathcal{G})) \tag{2}$$

During the evaluation stage, the test graph $\mathcal{G}_{\text{test}}$ is first transformed by *graph prompt* $g_\phi(\cdot)$, and the resulting prompted graph $g_\phi(\mathcal{G}_{\text{test}})$ is processed through the frozen GNN model $f$.

**Practical Usage.** In this part, we provide a detailed description of the refined process of *graph prompt tuning*, which comprises two fundamental steps: *template design* and *prompt optimization*.

*A. Template Design.* Given an input graph $\mathcal{G}$, we first generate a *graph template* $\mathcal{G}^*$, which includes learnable components in its adjacency matrix $\mathbf{A}^*$ and feature matrix $\mathbf{X}^*$. Previous research has attributed the success of prompt tuning to bridging the gap between pre-training tasks and downstream tasks [Liu et al., 2022a]. Consequently, it implies that the specific form of the graph template is influenced by the pre-training strategy employed by the model. For a specific pre-training task $t \in \mathbb{T}$ and an input graph $\mathcal{G}$, the graph template $\mathcal{G}^*$ can be expressed as:

$$\mathcal{G}^* \colon (\mathbf{A}^*, \mathbf{X}^*) = \psi_t(\mathcal{G}) \tag{3}$$

where the graph template $\mathcal{G}^*$ may contain learnable parameters (*i.e.*, tunable links or node features) in its adjacency matrix or feature matrix (similar to the inclusion of learnable soft prompts in a sentence), the candidate space for $\mathbf{A}^*$ is $\mathbb{A}$, and the candidate space for $\mathbf{X}^*$ is $\mathbb{X}$.

*B. Prompt Optimization.* Once we have obtained the graph template $\mathcal{G}^*$, our next step is to search for the optimal $\hat{\mathbf{A}}$ and $\hat{\mathbf{X}}$ within their respective candidate spaces $\mathbb{A}$ and $\mathbb{X}$ that maximize the likelihood of correctly predicting the labels $y$ using the pre-trained model $f$ and a learnable projection head $\theta$. This process can be expressed as:

$$\max_{\hat{\mathbf{A}}\in\mathbb{A},\hat{\mathbf{X}}\in\mathbb{X},\theta} P_{f,\theta}(y|\mathcal{G}^*) \tag{4}$$

The graph $\hat{\mathcal{G}}$ composed of $\hat{\mathbf{A}}$ and $\hat{\mathbf{X}}$ can be considered as the the *prompted graph* $g_\phi(\mathcal{G})$ mentioned in Formula 2.

**Practical Challenges.** The specific form of the graph template is closely tied to the pre-training task $t$ employed by the model $f$. However, designing the prompting function $\psi_t(\cdot)$ is challenging and varies for different pre-training tasks. Pioneering works [Sun et al., 2022, Liu et al., 2023] have proposed corresponding prompting functions $\psi_t(\cdot)$ for a specific pre-training strategy, with a focus on models pre-trained using edge prediction. However, many other pre-training strategies [Hu et al., 2020a, Xia et al., 2022b], such as attribute masking and context prediction, are widely utilized in existing pre-trained GNN models, yet no research has been conducted on designing prompting functions for these strategies. Furthermore, existing prompting functions are all intuitively designed, and these manual prompting functions lack a guarantee of effectiveness. It raises a natural question: *Can we design a universal prompting method that can be applied to any pre-trained model, regardless of the underlying pre-training strategy?*

### 3.3  Universal Graph Prompt Design

In this section, we introduce a universal prompting method and its variant. Drawing inspiration from the success of pixel-level Visual Prompt (VP) techniques [Bahng et al., 2022, Wu et al., 2022, Xing et al., 2022] in Computer Vision, our methods introduce learnable components to the feature space of the input graph. In Section 3.4, we will demonstrate that these prompting methods can theoretically achieve an equivalent effect as any prompting function $\psi_t(\cdot)$.

**Graph Prompt Feature (GPF).** GPF focuses on incorporating additional learnable parameters into the feature space of the input graph. Specifically, the learnable component $p$ is a vector of dimension $F$, where $F$ corresponds to the dimensionality of the node features. It can be denoted as:

$$p \in \mathbb{R}^F \tag{5}$$

The learnable vector $p$ is added to the graph features $\mathbf{X}$ to generate the prompted features $\mathbf{X}^*$, which can be expressed as:

$$\mathbf{X} = \{x_1, x_2, \ldots, x_N\} \quad \mathbf{X}^* = \{x_1 + p, x_2 + p, \ldots, x_N + p\} \tag{6}$$

The prompted features $\mathbf{X}^*$ replace the initial features $\mathbf{X}$ and are processed by the pre-trained model.

**Graph Prompt Feature-Plus (GPF-plus).** Building upon GPF, we introduce a variant called GPF-plus, which assigns an independent learnable vector $p_i$ to each node $v_i$ in the graph. It can be expressed as:

$$p_1, p_2, \ldots p_N \in \mathbb{R}^F \tag{7}$$

$$\mathbf{X} = \{x_1, x_2, \ldots, x_N\} \quad \mathbf{X}^* = \{x_1 + p_1, x_2 + p_2, \ldots, x_N + p_N\} \tag{8}$$

Similarly to GPF, the prompted features $\mathbf{X}^*$ replace the initial features $\mathbf{X}$ and are processed by the pre-trained model. However, such a design is not universally suitable for all scenarios. For instance, when training graphs have different scales (i.e., varying node numbers), it is challenging to train such a series of $p_i$. Additionally, when dealing with large-scale input graphs, such design requires a substantial amount of storage resources due to its $O(N)$ learnable parameters. To address these issues, we introduce an attention mechanism in the generation of $p_i$, making GPF-plus more parameter-efficient and capable of handling graphs with different scales. In practice, we train only $k$ independent basis vectors $p^b$, which can be expressed as:

$$p_1^b, p_2^b, \ldots p_k^b \in \mathbb{R}^F \tag{9}$$

where $k$ is a hyper-parameter that can be adjusted based on the downstream dataset. To obtain $p_i$ for node $v_i$, we utilize attentive aggregation of these basis vectors with the assistance of $k$ learnable linear projections $a$. The calculation process can be expressed as:

$$p_i = \sum_j^k \alpha_{i,j} p_j^b \qquad \alpha_{i,j} = \frac{\exp(a_j^{\mathrm{T}} x_i)}{\sum_l^k \exp(a_l^{\mathrm{T}} x_i)} \qquad (10)$$

Subsequently, $p_i$ is used to generate the prompted feature $\mathbf{X}^*$ as described in Formula 8.

## 3.4 Theoretical Analysis

In this section, we provide theoretical analyses for our proposed GPF and GPF-plus. Our analyses are divided into two parts. First, we certify the universality of our methods. We demonstrate that our approaches can theoretically achieve results equivalent to any prompting function $\psi_t(\cdot)$. It confirms the versatility and applicability of our methods across different pre-training strategies. Then, we make guarantee of the effectiveness of our proposed methods. Specifically, we demonstrate that our proposed graph prompt tuning is not weaker than full fine-tuning, which means that in certain scenarios, GPF and GPF-plus can achieve superior tuning results compared to fine-tuning. It is important to note that our derivations in the following sections are based on GPF, which adds a global extra vector $p$ to all nodes in the graph. GPF-plus, being a more powerful version, can be seen as an extension of GPF and degenerates to GPF when the hyperparameter $k$ is set to 1. Therefore, the analyses discussed for GPF are also applicable to GPF-plus.

Before we illustrate our conclusions, we first provide some preliminaries. For a given pre-training task $t \in \mathbb{T}$ and an input graph $\mathcal{G}: (\mathbf{A}, \mathbf{X})$, we assume the existence of a prompting function $\psi_t(\cdot)$ that generates a graph template $\mathcal{G}^*: (\mathbf{A}^*, \mathbf{X}^*) = \psi_t(\mathcal{G})$. The candidate space for $\mathbf{A}^*$ and $\mathbf{X}^*$ is denoted as $\mathbb{A}$ and $\mathbb{X}$, respectively.

**Theorem 1.** *(Universal Capability of GPF) Given a pre-trained GNN model $f$, an input graph $\mathcal{G}: (\mathbf{A}, \mathbf{X})$, an arbitrary prompting function $\psi_t(\cdot)$, for any prompted graph $\hat{\mathcal{G}}: (\hat{\mathbf{A}} \in \mathbb{A}, \hat{\mathbf{X}} \in \mathbb{X})$ in the candidate space of the graph template $\mathcal{G}^* = \psi_t(\mathcal{G})$, there exists a GPF extra feature vector $\hat{p}$ that satisfies:*

$$f(\mathbf{A}, \mathbf{X} + \hat{p}) = f(\hat{\mathbf{A}}, \hat{\mathbf{X}}) \qquad (11)$$

The complete proof of Theorem 1 can be found in the appendix. Theorem 1 implies that GPF can achieve the theoretical performance upper bound of any prompting function described in Formula 3 and 4. Specifically, if optimizing the graph template $\mathcal{G}^*$ generated by a certain prompting function $\psi_t(\cdot)$ can yield satisfactory graph representations, then theoretically, optimizing the vector $p$ of GPF can also achieve the exact graph representations. This conclusion may initially appear counter-intuitive since GPF only adds learnable components to node features without explicitly modifying the graph structure. The key lies in understanding that the feature matrix $\mathbf{X}$ and the adjacency matrix $\mathbf{A}$ are not entirely independent during the processing. The impact of graph structure modifications on the final graph representations can also be obtained through appropriate modifications to the node features. Therefore, GPF and GPF-plus, by avoiding the explicit illustration of the prompting function $\psi_t(\cdot)$, adopt a simple yet effective architecture that enables them to possess universal capabilities in dealing with pre-trained GNN models under various pre-training strategies.

Next, we make guarantee of the effectiveness of GPF and demonstrate that GPF is not weaker than fine-tuning, which means GPF can achieve better theoretical tuning results in certain situations compared to fine-tuning. In Natural Language Processing, the results obtained from fine-tuning are generally considered the upper bound for prompt tuning results [Lester et al., 2021a, Liu et al., 2021b, Ding et al., 2022]. It is intuitive to believe that fine-tuning, which allows for more flexible and comprehensive parameter adjustments in the pre-trained model, can lead to better theoretical results during downstream adaptation. However, in the graph domain, the architecture of graph neural networks magnifies the impact of input space transformation on the final representations to some extent. To further illustrate this point, following previous work [Kumar et al., 2022, Tian et al., 2023, Wei et al., 2021], we assume that the downstream task utilizes the squared regression loss $l = \sum_i (\hat{y}_i - y_i)^2$.

**Theorem 2.** *(Effectiveness Guarantee of GPF) For a pre-trained GNN model $f$, a series of graphs $\mathcal{D} = \{(\mathcal{G}_1: (\mathbf{A}_1, \mathbf{X}_1), y_1), \ldots, (\mathcal{G}_m: (\mathbf{A}_m, \mathbf{X}_m), y_m)\}$ under the non-degeneracy condition, and a*

*linear projection head $\theta$, there exists $\mathcal{Y}' = \{y_1', \ldots, y_m'\}$ for $y_1 = y_1', \ldots, y_m = y_m'$ that satisfies:*

$$l_{\text{GPF}} = \min_{p,\theta} \sum_i^m (f(\mathbf{A}_i, \mathbf{X}_i + p) \cdot \theta - y_i)^2 < l_{\text{FT}} = \min_{f,\theta} \sum_i^m (f(\mathbf{A}_i, \mathbf{X}_i) \cdot \theta - y_i)^2 \quad (12)$$

The detailed proof of Theorem 2 and the description of the degeneracy condition can be found in the appendix. Theorem 2 indicates that GPF obtains a lower minimum loss compared to fine-tuning in certain scenarios, demonstrating its ability to achieve better theoretical tuning results.

## 4 Experiments

### 4.1 Experiment Setup

**Model Architecture and Datasets.** We adopt the widely used 5-layer GIN [Xu et al., 2019] as the underlying architecture for our models, which aligns with the majority of existing pre-trained GNN models [Xia et al., 2022b, Hu et al., 2020a, Qiu et al., 2020a, You et al., 2020, Suresh et al., 2021, Xu et al., 2021b, Zhang et al., 2021b, You et al., 2022, Xia et al., 2022a]. As for the benchmark datasets, we employ the chemistry and biology datasets published by Hu et al. [2020a]. A comprehensive description of these datasets can be found in the appendix.

**Pre-training Strategies.** We employ five widely used strategies (tasks) to pre-train the GNN models, including Deep Graph Infomax (denoted by Infomax) [Velickovic et al., 2019a], Edge Prediction (denoted by EdgePred) [Kipf and Welling, 2016a], Attribute Masking (denoted by AttrMasking) [Hu et al., 2020a], Context Prediction (denoted by ContextPred) [Hu et al., 2020a] and Graph Contrastive Learning (denoted by GCL) [You et al., 2020]. A detailed description of these pre-training strategies can be found in the appendix.

**Tuning Strategies.** We adopt the pre-trained models to downstream tasks with different tuning strategies. Given a pre-trained GNN model $f$, a task-specific projection head $\theta$,

- *Fine Tuning* (denoted as FT). We tune the parameters of the pre-trained GNN model $f$ and the projection head $\theta$ simultaneously during the downstream training stage.

- *Graph Prompt Feature* (denoted as GPF). We freeze the parameters of the pre-trained model $f$ and introduce an extra learnable feature vector $p$ into the feature space of the input graph described as Formula 6. We tune the parameters of the projection head $\theta$ and feature vector $p$ during the downstream training stage.

- *Graph Prompt Feature-Plus* (denoted as GPF-plus). We freeze the parameters of pre-trained model $f$ and introduce $k$ learnable basis vectors $p_1^b, \ldots, p_k^b$ with $k$ learnable linear projections $a_1, \ldots, a_k$ to calculate the node-wise $p_i$ as Formula 10. We tune the parameters of the projection head $\theta$, basis vectors $p_1^b, \ldots, p_k^b$, and linear projections $a_1, \ldots, a_k$ during the downstream training stage.

**Implementation.** We perform five rounds of experiments with different random seeds for each experimental setting and report the average results. The projection head $\theta$ is selected from a range of [1, 2, 3]-layer MLPs with equal widths. The hyper-parameter $k$ of GPF-plus is chosen from the range [5,10,20]. Further details on the hyper-parameter settings can be found in the appendix.

### 4.2 Main Results

We compare the downstream performance of models trained using different pre-training and tuning strategies, and the overall results are summarized in Table 1. Our systematic study suggests the following observations:

*1. Our graph prompt tuning outperforms fine-tuning in most cases.* Based on the results presented in Table 1, it is evident that GPF and GPF-plus achieve superior performance compared to fine-tuning in the majority of cases. Specifically, GPF outperforms fine-tuning in 28/36 experiments, while GPF-plus outperforms fine-tuning in 29/36 experiments. It is worth noting that the tunable parameters in GPF and GPF-plus are significantly fewer in magnitude than those in fine-tuning (details can be found

Table 1: Test ROC-AUC (%) performance on molecular prediction benchmarks and protein function prediction benchmarks with different pre-training strategies and different tuning strategies.

| Pre-training Strategy | Tuning Strategy | BBBP | Tox21 | ToxCast | SIDER | ClinTox | MUV | HIV | BACE | PPI | **Avg.** |
|---|---|---|---|---|---|---|---|---|---|---|---|
| Infomax | FT | **67.55** ±2.06 | 78.57 ±0.51 | 65.16 ±0.53 | 63.34 ±0.45 | 70.06 ±1.45 | **81.42** ±2.65 | 77.71 ±0.45 | 81.32 ±1.25 | 71.29 ±1.79 | 72.93 |
| | GPF | 66.83 ±0.86 | 79.09 ±0.25 | 66.10 ±0.53 | **66.17** ±0.81 | 73.56 ±3.94 | 80.43 ±0.53 | 76.49 ±0.18 | 83.60 ±1.00 | 77.02 ±0.42 | 74.36 |
| | GPF-plus | 67.17 ±0.36 | **79.13** ±0.70 | **66.35** ±0.37 | 65.62 ±0.74 | **75.12** ±2.45 | 81.33 ±1.52 | **77.73** ±1.14 | **83.67** ±1.08 | **77.03** ±0.32 | **74.79** |
| AttrMasking | FT | 66.33 ±0.55 | 78.28 ±0.05 | 65.34 ±0.30 | 66.77 ±0.13 | 74.46 ±2.82 | 81.78 ±1.95 | 77.90 ±0.18 | 80.94 ±1.99 | 73.93 ±1.17 | 73.97 |
| | GPF | **68.09** ±0.38 | **79.04** ±0.90 | 66.32 ±0.42 | **69.13** ±1.16 | 75.06 ±1.02 | **82.17** ±0.65 | **78.86** ±1.42 | 84.33 ±0.54 | **78.91** ±0.25 | 75.76 |
| | GPF-plus | 67.71 ±0.64 | 78.87 ±0.31 | **66.58** ±0.13 | 68.65 ±0.72 | **76.17** ±2.98 | 81.12 ±1.32 | 78.13 ±1.12 | **85.76** ±0.36 | 78.90 ±0.11 | **75.76** |
| ContextPred | FT | **69.65** ±0.87 | 78.29 ±0.44 | 66.39 ±0.57 | 64.45 ±0.6 | 73.71 ±1.57 | 82.36 ±1.22 | **79.20** ±0.51 | 84.66 ±0.84 | 72.10 ±1.94 | 74.53 |
| | GPF | 68.48 ±0.88 | 79.99 ±0.24 | **67.92** ±0.35 | 66.18 ±0.46 | 74.51 ±2.72 | 84.34 ±0.25 | 78.62 ±1.46 | 85.32 ±0.41 | 77.42 ±0.07 | 75.86 |
| | GPF-plus | 69.15 ±0.82 | **80.05** ±0.46 | 67.58 ±0.54 | **66.94** ±0.95 | **75.25** ±1.88 | **84.48** ±0.78 | 78.40 ±0.16 | **85.81** ±0.43 | 77.71 ±0.21 | **76.15** |
| GCL | FT | 69.49 ±0.35 | 73.35 ±0.70 | 62.54 ±0.26 | 60.63 ±1.26 | **75.17** ±2.14 | 69.78 ±1.44 | **78.26** ±0.73 | 75.51 ±2.01 | 67.76 ±0.78 | 70.27 |
| | GPF | 71.11 ±1.20 | **73.64** ±0.25 | 62.70 ±0.46 | 61.26 ±0.53 | 72.06 ±2.98 | 70.09 ±0.67 | 75.52 ±1.09 | 78.55 ±0.56 | 67.60 ±0.57 | 70.28 |
| | GPF-plus | **72.18** ±0.93 | 73.35 ±0.43 | **62.76** ±0.75 | **62.37** ±0.38 | 73.90 ±2.47 | **72.94** ±1.87 | 77.51 ±0.82 | **79.61** ±2.06 | 67.89 ±0.69 | **71.39** |

Table 2: Test ROC-AUC (%) performance on molecular prediction benchmarks and protein function prediction benchmarks for the models pre-trained by Edge Prediction.

| Pre-training Strategy | Tuning Strategy | BBBP | Tox21 | ToxCast | SIDER | ClinTox | MUV | HIV | BACE | PPI | **Avg.** |
|---|---|---|---|---|---|---|---|---|---|---|---|
| EdgePred | FT | 66.56 ±3.56 | 78.67 ±0.35 | **66.29** ±0.45 | 64.35 ±0.78 | 69.07 ±4.61 | 79.67 ±1.70 | 77.44 ±0.58 | 80.90 ±0.92 | 71.54 ±0.85 | 72.72 |
| | GPPT | 64.13 ±0.14 | 66.41 ±0.04 | 60.34 ±0.14 | 54.86 ±0.25 | 59.81 ±0.46 | 63.05 ±0.34 | 60.54 ±0.54 | 70.85 ±1.42 | 56.23 ±0.27 | 61.80 |
| | GPPT (w/o ol) | 69.43 ±0.18 | 78.91 ±0.15 | 64.86 ±0.11 | 60.94 ±0.18 | 62.15 ±0.69 | 82.06 ±0.53 | 73.19 ±0.19 | 70.31 ±0.99 | 76.85 ±0.26 | 70.97 |
| | GraphPrompt | 69.29 ±0.19 | 68.09 ±0.19 | 60.54 ±0.21 | 58.71 ±0.13 | 55.37 ±0.57 | 62.35 ±0.44 | 59.31 ±0.93 | 67.70 ±1.26 | 49.48 ±0.96 | 61.20 |
| | GPF | **69.57** ±0.21 | 79.74 ±0.03 | 65.65 ±0.30 | 67.20 ±0.99 | **69.49** ±5.17 | 82.86 ±0.23 | 77.60 ±1.45 | 81.57 ±1.08 | 76.98 ±0.20 | 74.51 |
| | GPF-plus | 69.06 ±0.68 | **80.04** ±0.06 | 65.94 ±0.31 | **67.51** ±0.59 | 68.80 ±2.58 | **83.13** ±0.42 | **77.65** ±1.90 | **81.75** ±2.09 | **77.00** ±0.12 | **74.54** |

in the appendix). These experimental findings highlight the efficacy of our methods and demonstrate their capability to unleash the power of the pre-trained models.

*2. GPF and GPF-plus exhibit universal capability across various pre-training strategies.* GPF and GPF-plus present favorable tuning performance across all pre-training strategies examined in our experiments, consistently surpassing the average results obtained from fine-tuning. Specifically, GPF achieves an average improvement of 1.14%, while GPF-plus achieves an average improvement of 1.60%. These results signify the universal capability of GPF and GPF-plus, enabling their application to models trained with any pre-training strategy.

*3. GPF-plus marginally outperforms GPF.* Among the two graph prompt tuning methods, GPF-plus performs better than GPF in the majority of experiments (26/36). As discussed in Section 3.3, GPF-plus offers greater flexibility and expressiveness compared to GPF. The results further affirm that GPF-plus is an enhanced version of graph prompt tuning, aligning with the theoretical analysis.

## 4.3 Comparison with Existing Graph Prompt-based Methods

We also conducted a comparative analysis between our proposed methods, GPF and GPF-plus, and existing graph prompt-based tuning approaches [Sun et al., 2022, Liu et al., 2023]. Both of them

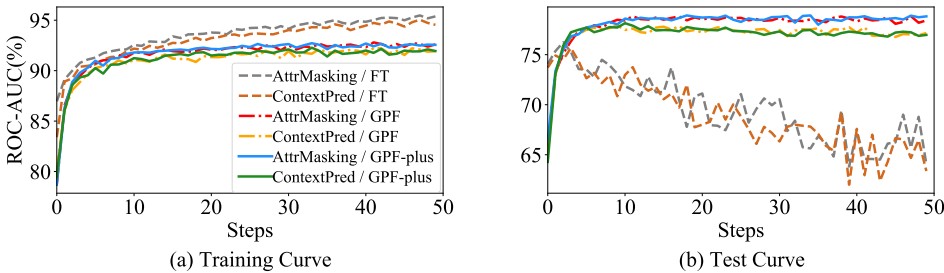

(a) Training Curve              (b) Test Curve

Figure 2: Training and test curves of different tuning methods.

specialize in tuning the models pre-trained by Edge Prediction (also known as Link Prediction). We apply GPPT [Sun et al., 2022], GPPT without orthogonal prompt constraint loss (denoted as GPPT (w/o ol)) [Sun et al., 2022], GraphPrompt [Liu et al., 2023] to the models pre-trained using Edge Prediction, and the results are summarized in Table 2. It is worth mentioning that GPPT is originally designed for node classification tasks. Therefore, we make minor modifications by substituting class-prototype nodes with class-prototype graphs to adapt it for graph classification tasks. The experimental results indicate that our proposed GPF and GPF-plus outperform existing graph prompt-based tuning methods by a significant margin. On the chemistry and biology benchmarks, GPF and GPF-plus achieve average improvements of 12%, 3%, and 13% over GPPT, GPPT (w/o ol), and GraphPrompt, respectively. These results showcase the ability of GPF and GPF-plus to achieve superior results compared to existing graph prompt-based tuning methods designed specifically for the pre-training strategy. Furthermore, it is worth highlighting that GPF and GPF-plus are the only two graph prompt-based tuning methods that surpass the performance of fine-tuning.

### 4.4 Additional Experiments

**Few-shot graph classification.** Prompt tuning has also been recognized for its effectiveness in addressing few-shot downstream tasks [Brown et al., 2020, Schick and Schütze, 2020b,a, Liu et al., 2021d, 2023]. We evaluate the efficacy of our proposed methods in handling few-shot scenarios. To conduct few-shot graph classification on the chemistry and biology datasets, we limit the number of training samples in the downstream tasks to 50 (compared to the original range of 1.2k to 72k training samples). The results are summarized in Table 4 of the appendix. Compared to the full-shot scenarios, our proposed graph prompt tuning demonstrates even more remarkable performance improvement (an average improvement of 2.95% for GPF and 3.42% for GPF-plus) over fine-tuning in the few-shot scenarios. This finding indicates that our solutions retain a higher degree of generalization ability in pre-trained models during few-shot downstream adaptations compared to fine-tuning.

**Training process analysis.** We conducted an analysis of the training process using different tuning methods on the biology datasets with the GNN models that employ Attribute Masking and Context Prediction as their pre-training tasks [Hu et al., 2020b]. Figure 2 presents the training and test curves during the adaptation stage. From Figure 2 (a), it can be observed that the ROC-AUC scores of the training set consistently increase during the adaptation stage for both our proposed methods and fine-tuning. However, from Figure 2 (b), we can find that their behavior on the test set is quite distinct. For fine-tuning, the ROC-AUC scores on the test set exhibit fluctuations and continuously decrease after an initial increase. On the other hand, when applying GPF or GPF-plus to adapt pre-trained models, the ROC-AUC scores on the test set continue to grow and remain consistently high. These results indicate that fully fine-tuning a pre-trained GNN model on a downstream task may lose the model's generalization ability. In contrast, employing our proposed graph prompt tuning methods can significantly alleviate this issue and maintain superior performance on the test set.

## 5 Conclusion

In this paper, we introduce a universal prompt-based tuning method for pre-trained GNN models. Our method GPF and its variant GPF-plus operate on the feature space of the downstream input graph. GPF and GPF-plus can theoretically achieve an equivalent effect to any form of prompting function, meaning we no longer need to illustrate the prompting function corresponding to each pre-training strategy explicitly. Instead, we can adaptively use GPF to obtain the prompted graph for downstream task adaptation. Compared to fine-tuning, the superiority of our method is demonstrated both theoretically and empirically, making it a compelling alternative for downstream adaptations.

# 6 Acknowledgements

This work was partially supported by Zhejiang NSF (LR22F020005), the National Key Research and Development Project of China (2018AAA0101900), and the Fundamental Research Funds for the Central Universities.

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

# A  Extra Materials for Section 3

## A.1  Extension to node-wise tasks

In this section, we illustrate the process of graph prompt tuning for node-wise tasks (node classification and link prediction). These tasks differ from graph classification, requiring node-level representations instead of graph-level representations. To bridge this gap, we employ *Subgraph GNNs* [Cotta et al., 2021, Zhang and Li, 2021, Bevilacqua et al., 2022, Zhao et al., 2022] to capture the graph-level and node-level representations. Subgraph GNN models utilize MPNNs on sets of subgraphs extracted from the original input graph. They subsequently aggregate the resulting representations [Frasca et al., 2022]. As a result, the node representations can also be interpreted as graph representations of the induced subgraphs. Specifically, the node representation $h_i$ for node $v_i$ can be calculated as:

$$h_i = f(\mathcal{G}_i) \tag{13}$$

where $f$ is a Subgraph GNN model, and $\mathcal{G}_i$ is the induced subgraph for node $v_i$. To obtain the node representations through graph prompt tuning, we freeze the model $f$ and introduce a learnable graph prompt $g_\phi \colon \mathbb{G} \to \mathbb{G}$, parameterized by $\phi$, as indicated in Formula 2, which can be expressed as:

$$h_i = f(g_\phi(\mathcal{G}_i)) \tag{14}$$

Once we acquire the node representations, we can seamlessly proceed with downstream node classification or link prediction tasks. Our proposed methods, GPF and GPF-plus, enable the effective execution of node-level tasks through the approach described in Formula 14.

## A.2  Proof for Theorem 1

**Theorem 1.**  *(Universal Capability of GPF) Given a pre-trained GNN model $f$, an input graph $\mathcal{G} \colon (\mathbf{A}, \mathbf{X})$, an arbitrary prompting function $\psi_t(\cdot)$, for any prompted graph $\hat{\mathcal{G}} \colon (\hat{\mathbf{A}} \in \mathbb{A}, \hat{\mathbf{X}} \in \mathbb{X})$ in the candidate space of the graph template $\mathcal{G}^* = \psi_t(\mathcal{G})$, there exists a GPF extra feature vector $\hat{p}$ that satisfies:*

$$f(\mathbf{A}, \mathbf{X} + \hat{p}) = f(\hat{\mathbf{A}}, \hat{\mathbf{X}})$$

*where $\psi_t(\mathcal{G}) = \mathcal{G}^* \colon (\mathbf{A}^*, \mathbf{X}^*)$, $\mathbb{A}$ and $\mathbb{X}$ are the candidate space for $\mathbf{A}^*$ and $\mathbf{X}^*$ respectively.*

To illustrate Theorem 1, we provide the specific architecture of the pre-trained GNN model $f$. For analytical simplicity, we initially assume that $f$ is a single-layer GIN [Xu et al., 2019] with a linear transformation. Subsequently, we extend our conclusions to multi-layer models utilizing various transition matrices [Klicpera et al., 2019]. During the generation of graph representations, we first obtain node representations and then employ a readout function to calculate the final graph representations. Existing pre-trained GNN models commonly employ *sum* or *mean* pooling for this purpose. Previous research [Mesquita et al., 2020] suggests that complex pooling mechanisms are unnecessary, as simple readout functions can yield superior performance. It is worth mentioning that the subsequent derivations still hold when we use any other weighted aggregation readout functions, such as average pooling, min/max pooling, and hierarchical pooling. Hence, we assume that the concrete architecture of the pre-trained GNN model $f$ can be expressed as:

$$\mathbf{H} = (\mathbf{A} + (1 + \epsilon) \cdot \mathbf{I}) \cdot \mathbf{X} \cdot \mathbf{W} \tag{15}$$

$$h_{\mathcal{G}} = \sum_{v_i \in \mathcal{V}} h_i \tag{16}$$

where $\mathbf{W}$ is a linear projection. The parameters $\epsilon$ and $\mathbf{W}$ have been pre-trained in advance and remain fixed during downstream adaptation. Next, we proceed with the derivation of Theorem 1. In Theorem 1, we impose no constraints on the form of the prompting function $\psi_t(\cdot)$, allowing $\hat{\mathbf{A}}$ and $\hat{\mathbf{X}}$ to represent the adjacency matrix and feature matrix of any graph. We define the graph-level transformation $g \colon \mathbb{G} \to \mathbb{G}$, which satisfies $(\hat{\mathbf{A}}, \hat{\mathbf{X}}) = g(\mathbf{A}, \mathbf{X})$. Consequently, Theorem 1 is equivalent to Proposition 1.

**Proposition 1.**  *Given a pre-trained GNN model $f$, an input graph $\mathcal{G} \colon (\mathbf{A}, \mathbf{X})$, for any graph-level transformation $g \colon \mathbb{G} \to \mathbb{G}$, there exists a GPF extra feature vector $\hat{p}$ that satisfies:*

$$f(\mathbf{A}, \mathbf{X} + \hat{p}) = f(g(\mathbf{A}, \mathbf{X})) \tag{17}$$

To further illustrate the graph-level transformation $g(\cdot)$, we decompose it into several specific transformations.

**Proposition 2.** *Given an input graph $\mathcal{G} = \{\mathbf{A}, \mathbf{X}\}$, an arbitrary graph-level transformation $g\colon \mathbb{G} \to \mathbb{G}$ can be decoupled to a series of following transformations:*

- ***Feature transformations.*** *Modifying the node features and generating the new feature matrix $\mathbf{X}' = g_{ft}(\mathbf{X})$.*

- ***Link transformations.*** *Adding or removing edges and generating the new adjacency matrix $\mathbf{A}' = g_{lt}(\mathbf{A})$.*

- ***Isolated component transformations.*** *Adding or removing isolated components (sub-graphs) and generating the new adjacency matrix and feature matrix $\mathbf{A}', \mathbf{X}' = g_{ict}(\mathbf{A}, \mathbf{X})$.*

Here, the word "isolated" refers to a component (sub-graph) that does not link with the rest of the graph. Proposition 2 suggests that an arbitrary graph-level transformation is a combination of the three transformations mentioned above. For instance, the deletion of a node in the initial graph can be decomposed into two steps: removing its connected edges and then removing the isolated node.

**Proposition 3.** *Given a pre-trained GNN model $f$, an input graph $\mathcal{G}\colon (\mathbf{A}, \mathbf{X})$, for any feature transformation $g_{ft}\colon g_{ft}(\mathbf{X}) = \mathbf{X}'$, there exists a GPF extra feature vector $\hat{p}$ that satisfies:*

$$f(\mathbf{A}, \mathbf{X} + \hat{p}) = f(\mathbf{A}, \mathbf{X}') \tag{18}$$

*Proof.* We set $\Delta \mathbf{X} = \mathbf{X}' - \mathbf{X}$. Then, we have:

$$\mathbf{H}' = (\mathbf{A} + (1 + \epsilon) \cdot \mathbf{I}) \cdot \mathbf{X}' \cdot \mathbf{W} \tag{19}$$

$$= (\mathbf{A} + (1 + \epsilon) \cdot \mathbf{I}) \cdot (\mathbf{X} + \Delta \mathbf{X}) \cdot \mathbf{W} \tag{20}$$

$$= (\mathbf{A} + (1 + \epsilon) \cdot \mathbf{I}) \cdot \mathbf{X} \cdot \mathbf{W} + (\mathbf{A} + (1 + \epsilon) \cdot \mathbf{I}) \cdot \Delta \mathbf{X} \cdot \mathbf{W} \tag{21}$$

$$= \mathbf{H} + (\mathbf{A} + (1 + \epsilon) \cdot \mathbf{I}) \cdot \Delta \mathbf{X} \cdot \mathbf{W} \tag{22}$$

For GPF $p = [\alpha_1, \cdots, \alpha_F] \in \mathbb{R}^{1 \times F}$, we can perform a similar split:

$$\mathbf{H}_p = (\mathbf{A} + (1 + \epsilon) \cdot \mathbf{I}) \cdot (\mathbf{X} + [1]^N \cdot p) \cdot \mathbf{W} \tag{23}$$

$$= (\mathbf{A} + (1 + \epsilon) \cdot \mathbf{I}) \cdot \mathbf{X} \cdot \mathbf{W} + (\mathbf{A} + (1 + \epsilon) \cdot \mathbf{I}) \cdot [1]^N \cdot p \cdot \mathbf{W} \tag{24}$$

$$= \mathbf{H} + (\mathbf{A} + (1 + \epsilon) \cdot \mathbf{I}) \cdot [1]^N \cdot p \cdot \mathbf{W} \tag{25}$$

$$= \mathbf{H} + [d_i + 1 + \epsilon]^N \cdot p \cdot \mathbf{W} \tag{26}$$

where $[1]^N \in \mathbb{R}^{N \times 1}$ denotes a column vector with $N$ 1s, $[d_i + 1 + \epsilon]^N \in \mathbb{R}^{N \times 1}$ denotes a column vector with the value of $i$-th row is $d_i + 1 + \epsilon$ and $d_i$ represents the degree number of $v_i$. To obtain the same graph representation $h_{\mathcal{G}}$, we have:

$$h_{\mathcal{G},ft} = h_{\mathcal{G},p} \to \mathrm{Sum}(\mathbf{H}') = \mathrm{Sum}(\mathbf{H}_p) \tag{27}$$

where $\mathrm{Sum}(\mathbf{M}) = \sum_i m_i$ denotes the operation that calculates the sum vector for each row in the matrix. We can further simplify the above equation as:

$$h_{\mathcal{G},ft} = h_{\mathcal{G},p} \tag{28}$$

$$\to \mathrm{Sum}(\mathbf{H}') = \mathrm{Sum}(\mathbf{H}_p) \tag{29}$$

$$\to \mathrm{Sum}(\mathbf{H} + (\mathbf{A} + (1 + \epsilon) \cdot \mathbf{I}) \cdot \Delta \mathbf{X} \cdot \mathbf{W}) = \mathrm{Sum}(\mathbf{H} + [d_i + 1 + \epsilon]^N \cdot p \cdot \mathbf{W}) \tag{30}$$

$$\to \mathrm{Sum}((\mathbf{A} + (1 + \epsilon) \cdot \mathbf{I}) \cdot \Delta \mathbf{X} \cdot \mathbf{W}) = \mathrm{Sum}([d_i + 1 + \epsilon]^N \cdot p \cdot \mathbf{W}) \tag{31}$$

where the results of $((\mathbf{A} + (1 + \epsilon) \cdot \mathbf{I}) \cdot \Delta \mathbf{X}) \in \mathbb{R}^{N \times F}$, and the frozen linear transformation $\mathbf{W} \in \mathbb{R}^{F \times F'}$. We first calculate $\Delta h_{\mathcal{G},p} = \mathrm{Sum}([d_i + 1 + \epsilon]^N \cdot p \cdot \mathbf{W}) \in \mathbb{R}^{F'}$. We can obtain that:

$$\Delta h_{\mathcal{G},p}^i = \sum_{j=1}^{F} \sum_{k=1}^{N} (d_k + 1 + \epsilon) \cdot \alpha_j \cdot \mathbf{W}_{j,i} \tag{32}$$

$$= \sum_{j=1}^{F} (D + N + N \cdot \epsilon) \cdot \alpha_j \cdot \mathbf{W}_{j,i} \tag{33}$$

where $h^i_{\mathcal{G},p}$ denotes the value of the $i$-th dimension in $h_{\mathcal{G},p}$, $D = \sum_{k=1}^{N} d_k$ denotes the total degree of all nodes in the whole graph, $\alpha_j, j \in [1, F]$ denotes the $j$-th learnable parameter in GPF $p$, and $\mathbf{W}_{j,i}, j \in [1, F], i \in [1, F']$ denotes the frozen parameter in $\mathbf{W}$. As for $\Delta h_{\mathcal{G},ft} = \text{Sum}((\mathbf{A} + (1 + \epsilon) \cdot \mathbf{I}) \cdot \Delta \mathbf{X} \cdot \mathbf{W})$, we assume $(\mathbf{A} + (1 + \epsilon) \cdot \mathbf{I}) \cdot \Delta \mathbf{X} = \mathbf{B} \in \mathbb{R}^{N \times F}$. Then we have:

$$\Delta h^i_{\mathcal{G},ft} = \sum_{j=1}^{F} (\sum_{k=1}^{N} \beta_{k,j}) \cdot \mathbf{W}_{j,i} \tag{34}$$

where $\beta_{k,j}, k \in [1, N], j \in [1, F]$ denotes the learnable parameter in $\mathbf{B}$. According to above analysis, to obtain a same graph representation $h_{\mathcal{G},\hat{p}}$ with a certain $h_{\mathcal{G},ft}$, we have:

$$h^i_{\mathcal{G},\hat{p}} = h^i_{\mathcal{G},ft} \text{ , for every } i \in [1, F'] \tag{35}$$

$$\rightarrow \Delta h^i_{\mathcal{G},\hat{p}} = \Delta h^i_{\mathcal{G},ft} \tag{36}$$

$$\rightarrow \alpha_j = \frac{\sum_{k=1}^{N} \beta_{k,j}}{D + N + N \cdot \epsilon} \text{ , } j \in [1, F] \tag{37}$$

Therefore, for an arbitrary feature transformation $g_{ft}$, there exists a GPF $\hat{p}$ that satisfies the above conditions and can obtain the exact graph representation for the pre-trained GNN model $f$. $\qquad \square$

Proposition 3 demonstrates the comprehensive coverage of our proposed GPF for all graph-level feature transformations. GPF introduces a uniform feature modification $p \in \mathbb{R}^F$ to each node in the graph. However, it can achieve an equivalent effect to adding independent feature modifications to each node individually under the pre-trained GNN model described above.

**Proposition 4.** *Given a pre-trained GNN model $f$, an input graph $\mathcal{G}$: $(\mathbf{A}, \mathbf{X})$, for any link transformation $g_{lt}$: $g_{lt}(\mathbf{A}) = \mathbf{A}'$, there exists a GPF extra feature vector $\hat{p}$ that satisfies:*

$$f(\mathbf{A}, \mathbf{X} + \hat{p}) = f(\mathbf{A}', \mathbf{X}) \tag{38}$$

*Proof.* The proof of Proposition 4 is similar to that of Proposition 3. We set $\Delta \mathbf{A} = \mathbf{A}' - \mathbf{A}$. It is worth mentioning that $\mathbf{A}, \mathbf{A}' \in \{0, 1\}^{N \times N}$ and $\Delta \mathbf{A} \in \{-1, 0, 1\}^{N \times N}$, which means they are of the same size, the values of $\mathbf{A}, \mathbf{A}'$ can only be 0 or 1, and the values of $\Delta \mathbf{A}$ can be $-1$, 0 or 1. We have:

$$\mathbf{H}' = (\mathbf{A}' + (1 + \epsilon) \cdot \mathbf{I}) \cdot \mathbf{X} \cdot \mathbf{W} \tag{39}$$

$$= ((\mathbf{A} + \Delta \mathbf{A}) + (1 + \epsilon) \cdot \mathbf{I}) \cdot \mathbf{X} \cdot \mathbf{W} \tag{40}$$

$$= \mathbf{H} + \Delta \mathbf{A} \cdot \mathbf{X} \cdot \mathbf{W} \tag{41}$$

From the proof of Proposition 3, we obtain:

$$\mathbf{H}_p = \mathbf{H} + [d_i + 1 + \epsilon]^N \cdot p \cdot \mathbf{W} \tag{42}$$

where $p = [\alpha_1, \cdots, \alpha_F] \in \mathbb{R}^{1 \times F}$ denotes our learnable GPF, $[d_i + 1 + \epsilon]^N \in \mathbb{R}^{N \times 1}$ denotes a column vector with the value of $i$-th line is $d_i + 1 + \epsilon$ and $d_i$ represents the degree number of $v_i$. With $\Delta h_{\mathcal{G},p} = \text{Sum}([d_i + 1 + \epsilon]^N \cdot p \cdot \mathbf{W}) \in \mathbb{R}^{F'}$, we can obtain:

$$\Delta h^i_{\mathcal{G},p} = \sum_{j=1}^{F} \sum_{k=1}^{N} (d_k + 1 + \epsilon) \cdot \alpha_j \cdot \mathbf{W}_{j,i} \tag{43}$$

$$= \sum_{j=1}^{F} (D + N + N \cdot \epsilon) \cdot \alpha_j \cdot \mathbf{W}_{j,i} \tag{44}$$

where $h^i_{\mathcal{G},p}$ denotes the value of the $i$-th dimension in $h_{\mathcal{G},p}$, $D = \sum_{k=1}^{N} d_k$ denotes the total degree of all nodes in the whole graph, $\alpha_j, j \in [1, F]$ denotes the $j$-th learnable parameter in GPF $p$, and $\mathbf{W}_{j,i}, j \in [1, F], i \in [1, F']$ denotes the frozen parameter in $\mathbf{W}$. As for $\Delta h_{\mathcal{G},lt} = \text{Sum}(\Delta \mathbf{A} \cdot \mathbf{X} \cdot \mathbf{W})$, we have:

$$\Delta h^i_{\mathcal{G},lt} = \sum_{j=1}^{F} \sum_{(k,l) \in N \times N} (\Delta a_{k,l} \cdot x_{l,j}) \cdot \mathbf{W}_{j,i} \tag{45}$$

where $\Delta a_{k,l}, k \in [1, N], l \in [1, N]$ denotes the element of $\Delta\mathbf{A}$, and $x_{l,j}, l \in [1, N], j \in [1, F]$ denotes the element of $\mathbf{X}$. To obtain a same graph representation $h_{\mathcal{G},\hat{p}}$ with a certain $h_{\mathcal{G},lt}$, we have:

$$h^i_{\mathcal{G},\hat{p}} = h^i_{\mathcal{G},lt} \text{ , for every } i \in [1, F'] \tag{46}$$

$$\rightarrow \Delta h^i_{\mathcal{G},\hat{p}} = \Delta h^i_{\mathcal{G},lt} \tag{47}$$

$$\rightarrow \alpha_j = \frac{\sum_{(k,l) \in N \times N} \Delta a_{k,l} \cdot x_{l,j}}{D + N + N \cdot \epsilon} \text{ , } j \in [1, F] \tag{48}$$

Therefore, for an arbitrary link transformation $g_{lt}$, there exists a GPF $\hat{p}$ that satisfies above conditions and can obtain the same graph representation for pre-trained GNN model $f$. $\qquad\square$

Proposition 4 demonstrates that GPF can also encompass all link transformations. Intuitively, link transformations are closely associated with changes in the adjacency matrix, which are independent of node features. However, our findings reveal that, within the architecture of most existing GNN models, modifications in the feature space and modifications in the structural space can produce equivalent effects.

**Proposition 5.** *Given a pre-trained GNN model $f$, an input graph $\mathcal{G}: (\mathbf{A}, \mathbf{X})$, for any isolated component transformation $g_{ict}: g_{ict}(\mathbf{A}, \mathbf{X}) = \mathbf{A}', \mathbf{X}'$, there exists a GPF extra feature vector $\hat{p}$ that satisfies:*

$$f(\mathbf{A}, \mathbf{X} + \hat{p}) = f(\mathbf{A}', \mathbf{X}') \tag{49}$$

*Proof.* Unlike feature transformations and linear transformations, isolated component transformations will change the number of nodes in the graph, which means the scale of modified $\mathbf{A}'$ and $\mathbf{X}'$ is uncertain. We first express the isolated component transformation in more details. The adjacency matrix $\mathbf{A}$ and feature matrix $\mathbf{X}$ can be divided into several isolated components, which can be expressed as:

$$\mathbf{A} = \begin{pmatrix} \mathbf{A}_1 & 0 & \cdots & 0 \\ 0 & \mathbf{A}_2 & \cdots & 0 \\ \vdots & \vdots & & \vdots \\ 0 & 0 & \cdots & \mathbf{A}_m \end{pmatrix} \qquad \mathbf{X} = \begin{pmatrix} \mathbf{X}_1 \\ \mathbf{X}_2 \\ \vdots \\ \mathbf{X}_m \end{pmatrix} \tag{50}$$

Removing an isolated component $\mathcal{C}_k = \{\mathbf{A}_k, \mathbf{X}_k\}, k \in [1, m]$ means removing both $\mathbf{A}_k$ in the adjacency matrix and corresponding $\mathbf{X}_k$ in the feature matrix. Adding a new isolated component $\mathcal{C}_{m+l} = \{\mathbf{A}_{m+l}, \mathbf{X}_{m+l}\}, l \geq 1$ means adding $\mathbf{A}_{m+l}$ to the adjacency matrix $\mathbf{A}$, and adding $\mathbf{X}_{m+l}$ to the corresponding position of $\mathbf{X}$. Then we have:

$$h_{\mathcal{G},ist} = \sum_k \text{Sum}((\mathbf{A}_k + (1 + \epsilon) \cdot \mathbf{I}) \cdot \mathbf{X}_k \cdot \mathbf{W}) \tag{51}$$

To align with the proofs of Proposition 3 and Proposition 4, we set $\Delta h_{\mathcal{G},ist} = h_{\mathcal{G},ist} - \text{Sum}((\mathbf{A} + (1 + \epsilon) \cdot \mathbf{I}) \cdot \mathbf{X} \cdot \mathbf{W})$, and it can be expressed as:

$$\Delta h_{\mathcal{G},ist} = \sum_k I_k \cdot \text{Sum}((\mathbf{A}_k + (1 + \epsilon) \cdot \mathbf{I}) \cdot \mathbf{X}_k \cdot \mathbf{W}) \tag{52}$$

where $I_k$ is an indicator that satisfies:

$$I_k = \begin{cases} 0 & \text{if } \mathcal{C}_k \text{ has no change} \\ 1 & \text{if } \mathcal{C}_k \text{ is an additional component} \\ -1 & \text{if } \mathcal{C}_k \text{ is a removed component} \end{cases} \tag{53}$$

From the proof of Proposition 3, we have following conclusions:

$$\mathbf{H}_p = \mathbf{H} + [d_i + 1 + \epsilon]^N \cdot p \cdot \mathbf{W} \tag{54}$$

where $p = [\alpha_1, \cdots, \alpha_F] \in \mathbb{R}^{1 \times F}$ denotes our learnable GPF, $[d_i + 1 + \epsilon]^N \in \mathbb{R}^{N \times 1}$ denotes a column vector with the value of $i$-th line is $d_i + 1 + \epsilon$ and $d_i$ represents the degree number of $v_i$.

With $\Delta h_{\mathcal{G},p} = \text{Sum}([d_i + 1 + \epsilon]^N \cdot p \cdot \mathbf{W}) \in \mathbb{R}^{F'}$, we can obtain:

$$\Delta h_{\mathcal{G},p}^i = \sum_{j=1}^{F} \sum_{k=1}^{N} (d_k + 1 + \epsilon) \cdot \alpha_j \cdot \mathbf{W}_{j,i} \tag{55}$$

$$= \sum_{j=1}^{F} (D + N + N \cdot \epsilon) \cdot \alpha_j \cdot \mathbf{W}_{j,i} \tag{56}$$

where $h_{\mathcal{G},p}^i$ denotes the value of the $i$-th dimension in $h_{\mathcal{G},p}$, $D = \sum_{k=1}^{N} d_k$ denotes the total degree of all nodes in the whole graph, $\alpha_j, j \in [1, F]$ denotes the $j$-th learnable parameter in GPF $p$, and $\mathbf{W}_{j,i}, j \in [1, F], i \in [1, F']$ denotes the frozen parameter in $\mathbf{W}$. To obtain a same graph representation $h_{\mathcal{G},\hat{p}}$ with a certain $h_{\mathcal{G},ist}$, we have:

$$h_{\mathcal{G},\hat{p}}^i = h_{\mathcal{G},ist}^i \text{ , for every } i \in [1, F'] \tag{57}$$

$$\rightarrow \Delta h_{\mathcal{G},\hat{p}}^i = \Delta h_{\mathcal{G},ist}^i \tag{58}$$

$$\rightarrow \alpha_j = \frac{\sum_k I_k \cdot \text{Sum}((\mathbf{A}_k + (1 + \epsilon) \cdot \mathbf{I}) \cdot \mathbf{X}_k^j)}{D + N + N \cdot \epsilon} \text{ , } j \in [1, F] \tag{59}$$

where $\mathbf{X}^j$ denotes the $j$-th column of the matrix $\mathbf{X}$. Therefore, for an arbitrary isolated component transformation $g_{ict}$, there exists a GPF $\hat{p}$ that satisfies above conditions and can obtain the same graph representation for pre-trained GNN model $f$. $\qquad\square$

The isolated component transformation possesses the capability to alter the scale of a graph, and previous studies have paid limited attention to this type of graph-level transformation.

**Proposition 6.** *Given a pre-trained GNN model $f$, an input graph $\mathcal{G}: (\mathbf{A}, \mathbf{X})$, for a series of transformations $\mathbf{g} = \{g_1, g_2, \cdots, g_k\}$ composed of $g_{ft}$, $g_{lt}$ and $g_{ist}$, there exists a GPF extra feature vector $\hat{p}$ that satisfies:*

$$f(\mathbf{A}, \mathbf{X} + \hat{p}) = f(\mathbf{g}(\mathbf{A}, \mathbf{X})) \tag{60}$$

*Proof.* Without loss of generality, we consider $\mathbf{g} = \{g_1, g_2\}$ with two transformations described in Proposition 2. Now we prove there exists a GPF $\hat{p}$ that satisfies:

$$f(\mathbf{A}, \mathbf{X} + p) = f(g_2(g_1(\mathbf{A}, \mathbf{X}))) \tag{61}$$

We assume $g_1(\mathbf{A}, \mathbf{X}) = \mathbf{A}', \mathbf{X}'$. According to Proposition 3, 4, 5, there exists a GPF $\hat{p}_1$ that satisfies:

$$f(\mathbf{A}, \mathbf{X} + \hat{p}_1) = f(g_1(\mathbf{A}, \mathbf{X})) \tag{62}$$

and a $\hat{p}_2$ that satisfies:

$$f(\mathbf{A}, \mathbf{X} + \hat{p}_2) = f(g_2(\mathbf{A}', \mathbf{X}')) \tag{63}$$

Therefore, there is a $\hat{p} = \hat{p}_1 + \hat{p}_2$ that satisfies:

$$f(\mathbf{A}, \mathbf{X} + \hat{p}) = f(g_2(g_1(\mathbf{A}, \mathbf{X}))) \tag{64}$$

$\qquad\square$

Based on the preceding analysis, we have established that the GPF can replicate the effects of any graph-level transformation on the pre-trained GNN model defined by Formula 15 and 16. Hence, we have successfully demonstrated Proposition 1 and Theorem 1 within the framework of the simple model architecture. Next, we aim to generalize our findings to more intricate scenarios.

**Extension to other GNN backbones.** We use GIN Xu et al. [2019] as the default backbone model in our previous derivation. When replacing GIN with another GNN model, only slight modifications are required to ensure that all propositions remain valid. As is described in Klicpera et al. [2019], various GNN models can be represented as:

$$\mathbf{H} = \mathbf{S} \cdot \mathbf{X} \cdot \mathbf{W} \tag{65}$$

where $\mathbf{S} \in \mathbb{R}^{N \times N}$ denotes the diffusion matrix (*e.g.*, $\mathbf{A} + (1 + \epsilon) \cdot \mathbf{I}$ is the diffusion matrix for GIN), and $\mathbf{W}$ denotes the linear projection. In this case, we modify the Formula 26, 42 and 54 as follows:

$$\mathbf{H}_p = \mathbf{H} + [d_i + 1 + \epsilon]^N \cdot p \cdot \mathbf{W} \tag{66}$$

$$\rightarrow \mathbf{H}_p = \mathbf{H} + [\sum_j s_{i,j}]^N \cdot p \cdot \mathbf{W} \tag{67}$$

where $s_{i,j}$ is the element of the diffusion matrix $\mathbf{S}$. With these modifications, Propositions 3, 4, and 5 remain valid.

**Extension to multi-layer models.** For analytical simplification, similar to many previous works, we consider multi-layer GNN models without non-linear activation functions between layers, which can be expressed as:

$$\mathbf{H}_{(0)} = \mathbf{X} \tag{68}$$

$$\mathbf{H}_{(k)} = \mathbf{S}_{(k)} \cdot \mathbf{H}_{(k-1)} \cdot \mathbf{W}_{(k)} \tag{69}$$

where $\mathbf{S}_{(k)}$ is the diffusion matrix described as Formula 65 of the $k$-th layer, and $\mathbf{W}_{(k)}$ is the linear projection of the $k$-th layer. With such architecture, the node representations of the $k$-th layer $\mathbf{H}_{(k)}$ can also be obtained as:

$$\mathbf{H}_{(k)} = (\prod_{i=1}^{k} \mathbf{S}_{(i)}) \cdot \mathbf{X} \cdot (\prod_{i=1}^{k} \mathbf{W}_{(i)}) = \mathbf{S}' \cdot \mathbf{X} \cdot \mathbf{W}' \tag{70}$$

where $\mathbf{S}' = \prod_{i=1}^{k} \mathbf{S}_{(i)}$ and $\mathbf{W}' = \prod_{i=1}^{k} \mathbf{W}_{(i)}$. By substituting $\mathbf{S} = \mathbf{S}'$ and $\mathbf{W} = \mathbf{W}'$ in Formula 65, we can find that Proposition 3, 4 and 5 still hold true.

### A.3 Proof for Theorem 2

**Theorem 2.** *(Effectiveness Guarantee of GPF) For a pre-trained GNN model $f$, a series of graphs $\mathcal{D} = \{(\mathcal{G}_1 : (\mathbf{A}_1, \mathbf{X}_1), y_1), \ldots, (\mathcal{G}_m : (\mathbf{A}_m, \mathbf{X}_m), y_m)\}$ under the non-degeneracy condition, and a linear projection head $\theta$, there exists $\mathcal{Y}' = \{y'_1, \ldots, y'_m\}$ for $y_1 = y'_1, \ldots, y_m = y'_m$ that satisfies:*

$$l_{\text{GPF}} = \min_{p,\theta} \sum_{i}^{m} (f(\mathbf{A}_i, \mathbf{X}_i + p) \cdot \theta - y_i)^2 < l_{\text{FT}} = \min_{f,\theta} \sum_{i}^{m} (f(\mathbf{A}_i, \mathbf{X}_i) \cdot \theta - y_i)^2$$

To demonstrate Theorem 2, we need to provide a more detailed description of the architecture of the GNN model $f$. We assume that the graph representations $h_{\mathcal{G}_i}$ for $\mathcal{G}_i$ are obtained through the following process:

$$\mathbf{H}_i = \mathbf{S}_i \cdot \mathbf{X}_i \cdot \mathbf{W} \tag{71}$$

$$h_{\mathcal{G}_i} = \text{Sum}(\mathbf{H}_i) \tag{72}$$

where $\mathbf{S}_i \in \mathbb{R}^{N_i \times N_i}$ denotes the diffusion matrix of $\mathcal{G}_i$ as Formula 65, $N_i$ denotes the node number of the graph $\mathcal{G}_i$, $\mathbf{W} \in \mathbb{R}^{F \times F'}$ denotes the linear projection, $F$ is the dimension of node features, and $\text{Sum}(\mathbf{M}) = \sum_i m_i$ denotes the operation that calculates the sum vector for each row in the matrix $\mathbf{M}$. When we employ our proposed GPF in the above model $f$, the graph representations $h_{\mathcal{G}_i}^{\text{GPF}}$ are calculated as:

$$\mathbf{H}_i^{\text{GPF}} = \mathbf{S}_i \cdot (\mathbf{X}_i + [1]^{\text{T}} \cdot p) \cdot \mathbf{W} \tag{73}$$

$$h_{\mathcal{G}_i}^{\text{GPF}} = \text{Sum}(\mathbf{H}_i^{\text{GPF}}) \tag{74}$$

where $[1]^{\text{T}}$ is a column vector with all 1's, and $p \in \mathbb{R}^{1 \times F}$ is the extra learnable vector of GPF. The squared regression loss $l$ can be expressed as:

$$l = \sum_{i}^{m} (h_{\mathcal{G}_i} \cdot \theta - y_i)^2 \tag{75}$$

where $\theta \in \mathbb{R}^{F' \times 1}$ is a linear projection head. In such case, the optimal tuning loss of fine-tuning $l_{\text{FT}}$ and GPF $l_{\text{GPF}}$ can be expressed as:

$$l_{\text{FT}} = \min_{\mathbf{W}, \theta} \sum_i^m (\text{Sum}(\mathbf{S}_i \cdot \mathbf{X}_i \cdot \mathbf{W}) \cdot \theta - y_i)^2 \tag{76}$$

$$l_{\text{GPF}} = \min_{p, \theta} \sum_i^m (\text{Sum}(\mathbf{S}_i \cdot (\mathbf{X}_i + [1]^{\text{T}} \cdot p) \cdot \mathbf{W}) \cdot \theta - y_i)^2 \tag{77}$$

Before we prove Theorem 2, we first illustrate the following proposition.

**Proposition 7.** *Given a series of graphs* $\mathcal{D} = \{\mathcal{G}_1 \colon (\mathbf{A}_1, \mathbf{X}_1), \ldots, \mathcal{G}_m \colon (\mathbf{A}_m, \mathbf{X}_m)\}$ *and a linear projection* $\hat{\mathbf{W}}$, *there exist* $\hat{p} \in \mathbb{R}^{1 \times F}$, $\hat{\theta} \in \mathbb{R}^{F' \times 1}$, *and* $\delta > 0$ *that satisfy:*

$$\sum_i^m \|\text{Sum}(\mathbf{S}_i \cdot (\mathbf{X}_i + [1]^{\text{T}} \cdot \hat{p}) \cdot \hat{\mathbf{W}}) \cdot \hat{\theta} - \text{Sum}(\mathbf{S}_i \cdot \mathbf{X}_i \cdot \mathbf{W}') \cdot \theta'\|_2 > \delta \tag{78}$$

*for any* $\mathbf{W}' \in \mathbb{R}^{F \times F'}$, $\theta' \in \mathbb{R}^{F' \times 1}$.

For analytical simplification, we gather all the unique node features in $\mathcal{D}$ into a set $\mathcal{S}_{\mathbf{X}} = \{x_1, \cdots, x_l\}$. Consequently, the node feature matrix $\mathbf{X}_i$ of any graph $\mathcal{G}i$ can be constructed from elements in the set $\mathcal{S}\mathbf{X}$. Next, we proceed to expand the function $\text{Sum}(\cdot)$ as:

$$h_{\mathcal{G}_i} = \text{Sum}(\mathbf{S}_i \cdot \mathbf{X}_i \cdot \mathbf{W}) = \sum_k^{N_i} \sum_j^{N_i} \mathbf{S}_{i,(j,k)} \cdot \mathbf{X}_{i,(k,:)} \cdot \mathbf{W} \tag{79}$$

$$h_{\mathcal{G}_i}^{\text{GPF}} = \text{Sum}(\mathbf{S}_i \cdot (\mathbf{X}_i + [1]^{\text{T}} \cdot \hat{p}) \cdot \mathbf{W})$$
$$= \sum_k^{N_i} \sum_j^{N_i} \mathbf{S}_{i,(j,k)} \cdot \mathbf{X}_{i,(k,:)} \cdot \mathbf{W} + \sum_k^{N_i} \sum_j^{N_i} \mathbf{S}_{i,(j,k)} \cdot p \cdot \mathbf{W} \tag{80}$$

where $\mathbf{S}_{i,(j,k)}$ denotes the element in the $j$-th row and $k$-th column of $\mathbf{S}_i$, and $\mathbf{X}_{i,(k,:)}$ denotes the $k$-th row vector of $\mathbf{X}_i$. In order to express the representations of different graphs in a unified form, we rewrite the above formulas to calculate the graph representations $h_{\mathcal{G}_i}$ and $h_{\mathcal{G}_i}^{\text{GPF}}$ from the node representations in $\mathcal{S}_{\mathbf{X}}$:

$$h_{\mathcal{G}_i} = \sum_k^{N_i} \sum_j^{N_i} \mathbf{S}_{i,(j,k)} \cdot \mathbf{X}_{i,(k,:)} \cdot \mathbf{W} = \sum_j^l c_{i,j} \cdot x_j \cdot \mathbf{W} \tag{81}$$

$$h_{\mathcal{G}_i}^{\text{GPF}} = \sum_k^{N_i} \sum_j^{N_i} \mathbf{S}_{i,(j,k)} \cdot \mathbf{X}_{i,(k,:)} \cdot \mathbf{W} + \sum_k^{N_i} \sum_j^{N_i} \mathbf{S}_{i,(j,k)} \cdot p \cdot \mathbf{W}$$

$$= \sum_j^l c_{i,j} \cdot x_j \cdot \mathbf{W} + \sum_j^l c_{i,j} \cdot p \cdot \mathbf{W} \tag{82}$$

where $c_{i,j}$ is the coefficient of $x_j$ for the graph $\mathcal{G}_i$, which is calculated as $c_{i,j} = \sum_k \sum_{j'}^{N_i} \mathbf{S}_{i,(j',k)}$ for all $k$'s that satisfy $\mathbf{X}_{i,(k,:)} = x_j$. We can also get that $\sum_j^l c_{i,j} = \sum_k^{N_i} \sum_{j'}^{N_i} \mathbf{S}_{i,(j',k)}$. Then, we rewrite Formula 78 into a matrix form as follows:

$$\|(\mathbf{C} \cdot \mathbf{X}_{\mathcal{S}_{\mathbf{X}}} + [\sum_j \mathbf{C}_{(i,j)}]^{\text{T}} \cdot \hat{p}) \cdot \hat{\mathbf{W}} \cdot \hat{\theta} - \mathbf{C} \cdot \mathbf{X}_{\mathcal{S}_{\mathbf{X}}} \cdot \mathbf{W}' \cdot \theta'\|_2 > \delta \tag{83}$$

where $\mathbf{X}_{\mathcal{S}_{\mathbf{X}}} \in \mathbb{R}^{l \times F}$ denotes the feature matrix for $\mathcal{S}_{\mathbf{X}}$ that satisfies $\mathbf{X}_{\mathcal{S}_{\mathbf{X}},(j,:)} = x_j$, $\mathbf{C} \in \mathbb{R}^{m \times l}$ denotes the coefficient matrix and the element in the $i$-th row and $k$-th column $\mathbf{C}_{(i,j)}$ is equal to $c_{i,j}$, $[\sum_j \mathbf{C}_{(i,j)}]^{\text{T}}$ denotes a column vector with the value of $i$-th row is $\sum_j \mathbf{C}_{(i,j)}$. We represent $\mathbf{W}' \cdot \theta' = \hat{\mathbf{W}} \cdot \hat{\theta} + \Delta\mathbf{W} \cdot \theta$, then we have:

$$\|(\mathbf{C} \cdot \mathbf{X}_{\mathcal{S}_{\mathbf{X}}} + [\sum_j \mathbf{C}_{(i,j)}]^{\text{T}} \cdot \hat{p}) \cdot \hat{\mathbf{W}} \cdot \hat{\theta} - \mathbf{C} \cdot \mathbf{X}_{\mathcal{S}_{\mathbf{X}}} \cdot (\hat{\mathbf{W}} \cdot \hat{\theta} + \Delta\mathbf{W} \cdot \theta)\|_2 \tag{84}$$

$$= \|[\sum_j \mathbf{C}_{(i,j)}]^{\text{T}} \cdot \hat{p} \cdot \hat{\mathbf{W}} \cdot \hat{\theta} - \mathbf{C} \cdot \mathbf{X}_{\mathcal{S}_{\mathbf{X}}} \cdot \Delta\mathbf{W} \cdot \theta\|_2 \tag{85}$$

As described by the condition of Proposition 7, $\mathbf{W}'$ and $\theta'$ can be chosen arbitrarily, which means $\Delta \mathbf{W} \cdot \theta$ can be equal to any $v \in \mathbb{R}^{F \times 1}$. Therefore, Proposition 7 can be reformed as below.

*Given a series of graphs $\mathcal{D} = \{\mathcal{G}_1 \colon (\mathbf{A}_1, \mathbf{X}_1), \ldots, \mathcal{G}_m \colon (\mathbf{A}_m, \mathbf{X}_m)\}$ and a linear projection $\hat{\mathbf{W}}$, there exist $\hat{p} \in \mathbb{R}^{1 \times F}$, $\hat{\theta} \in \mathbb{R}^{F' \times 1}$, and $\delta > 0$ that satisfy:*

$$\|[\sum_j \mathbf{C}_{(i,j)}]^T \cdot \hat{p} \cdot \hat{\mathbf{W}} \cdot \hat{\theta} - \mathbf{C} \cdot \mathbf{X}_{\mathcal{S}_{\mathbf{x}}} \cdot v\|_2 > \delta \tag{86}$$

*for any $v \in \mathbb{R}^{F \times 1}$.*

We make the assumption that $\mathbf{C}$ is a column full-rank matrix, which implies that there is no uniform feature distribution shared among different graphs, aligning with real-world scenarios. It is important to note that $\mathbf{W}$ is pre-trained beforehand, and for any $\hat{p}$ and $\hat{\theta}$ satisfying $\hat{p} \cdot \hat{\mathbf{W}} \cdot \hat{\theta} \neq 0$, the non-degeneracy condition for Formula 86 is as follows:

$$\|[\sum_j \mathbf{C}_{(i,j)}]^{\mathrm{T}} \cdot \hat{p} \cdot \hat{\mathbf{W}} \cdot \hat{\theta} - \mathbf{C} \cdot \mathbf{X}_{\mathcal{S}_{\mathbf{x}}} \cdot v\|_2 \neq 0 \tag{87}$$

$$\rightarrow \mathbf{X}_{\mathcal{S}_{\mathbf{x}}} \cdot v' = [1]^{\mathrm{T}} \text{ has no solution } v' \tag{88}$$

where $[1]^{\mathrm{T}} \in \mathbb{R}^{m \times 1}$ is a column vector with all 1's. Therefore, Formula 86 and Proposition 7 hold for all $\mathbf{X}_{\mathcal{S}_{\mathbf{x}}}$ for which $\mathbf{X}_{\mathcal{S}_{\mathbf{x}}} \cdot v' = [1]^{\mathrm{T}}$ has no solution $v'$.

Finally, we revisit Theorem 2. Given a series of graphs under the non-degeneracy condition, a pre-trained linear projection $\hat{\mathbf{W}}$, $\hat{p}$ and $\hat{\theta}$ that satisfy $\hat{p} \cdot \hat{\mathbf{W}} \cdot \hat{\theta} \neq 0$, we can construct $\mathcal{Y}' = \{y'_1, \ldots, y'_m\}$ as:

$$y'_i = \mathbf{C}_{(i,:)} \cdot \mathbf{X}_{\mathcal{S}_{\mathbf{x}}} \cdot \hat{\mathbf{W}} \cdot \hat{\theta} + \sum_j \mathbf{C}_{(i,j)} \cdot \hat{p} \cdot \hat{\mathbf{W}} \cdot \hat{\theta} \tag{89}$$

Under these conditions, the optimal theoretical tuning results for GPF and fine-tuning can be expressed as::

$$l_{\mathrm{GPF}} = \min_{p,\theta} \|(\mathbf{C} \cdot \mathbf{X}_{\mathcal{S}_{\mathbf{x}}} + [\sum_j \mathbf{C}_{(i,j)}]^{\mathrm{T}} \cdot p) \cdot \hat{\mathbf{W}} \cdot \theta - (\mathbf{C} \cdot \mathbf{X}_{\mathcal{S}_{\mathbf{x}}} + [\sum_j \mathbf{C}_{(i,j)}]^{\mathrm{T}} \cdot \hat{p}) \cdot \hat{\mathbf{W}} \cdot \hat{\theta}\|_2 = 0 \tag{90}$$

$$l_{\mathrm{FT}} = \min_{\mathbf{W},\theta} \|\mathbf{C} \cdot \mathbf{X}_{\mathcal{S}_{\mathbf{x}}} \cdot \mathbf{W} \cdot \theta - (\mathbf{C} \cdot \mathbf{X}_{\mathcal{S}_{\mathbf{x}}} + [\sum_j \mathbf{C}_{(i,j)}]^{\mathrm{T}} \cdot \hat{p}) \cdot \hat{\mathbf{W}} \cdot \hat{\theta}\|_2 > 0 \tag{91}$$

Here, $l_{\mathrm{GPF}}$ can be achieved with $p = \hat{p}$ and $\theta = \hat{\theta}$, while $l_{\mathrm{FT}}$ consistently remains greater than 0, as stated in Proposition 7. Therefore, we can conclude that $l_{\mathrm{GPF}} < l_{\mathrm{FT}}$, thus establishing the proof of Theorem 2.

# B  More Information on Experiments

## B.1  Details of the datasets

**Dataset overview**  We utilize the datasets provided by Hu et al. [2020a] for our pre-training datasets. These datasets consist of two domains: chemistry and biology. The chemistry domain dataset comprises 2 million unlabeled molecules sampled from the ZINC15 database [Sterling and Irwin, 2015], along with 256K labeled molecules obtained from the preprocessed ChEMBL dataset [Mayr et al., 2018, Gaulton et al., 2012]. On the other hand, the biology domain dataset includes 395K unlabeled protein ego-networks and 88K labeled protein ego-networks extracted from PPI networks. For the models pre-trained on the chemistry dataset, we employ eight binary graph classification datasets available in MoleculeNet [Wu et al., 2017] as downstream tasks. As for the models pre-trained on the biology dataset, we apply the pre-trained models to 40 binary classification tasks, with each task involving the prediction of a specific fine-grained biological function.

**Pre-training datasets.**  The datasets provided by Hu et al. [2020a] consist of two distinct datasets: Biology and Chemistry, corresponding to the biology domain and chemistry domain, respectively. The Biology dataset contains 395K unlabeled protein ego-networks obtained from PPI networks of 50 species. These networks are used for node-level self-supervised pre-training. Additionally, 88K labeled protein ego networks serve as the training data for predicting 5000 coarse-grained biological functions. This graph-level multi-task supervised pre-training aims to predict these functions jointly. Regarding the Chemistry dataset, it comprises 2 million unlabeled molecules sampled from the ZINC15 database [Sterling and Irwin, 2015]. These molecules are utilized for node-level self-supervised pre-training. For graph-level multi-task supervised pre-training, a preprocessed ChEMBL dataset [Mayr et al., 2018, Gaulton et al., 2012] is employed. This dataset contains 456K molecules and covers 1310 different biochemical assays.

**Downstream datasets.**  The statistics of the downstream datasets utilized for the models pre-trained on Biology and Chemistry are presented in Table 3.

Table 3: Statistics of datasets for downstream tasks.

| Dataset | BBBP | Tox21 | ToxCast | SIDER | ClinTox | MUV | HIV | BACE | PPI |
|---|---|---|---|---|---|---|---|---|---|
| # Proteins / Molecules | 2039 | 7831 | 8575 | 1427 | 1478 | 93087 | 41127 | 1513 | 88K |
| # Binary prediction tasks | 1 | 12 | 617 | 27 | 2 | 17 | 1 | 1 | 40 |

## B.2  Details of pre-training strategies

We adopt five widely used strategies (tasks) to pre-train the GNN models, which are listed as below:

- *Deep Graph Infomax* (denoted by Infomax). It is first proposed by Velickovic et al. [2019a]. Deep graph infomax obtains expressive representations for graphs or nodes via maximizing the mutual information between graph-level representations and substructure-level representations of different granularity.

- *Edge Prediction* (denoted by EdgePred). It is a regular graph reconstruction task used by many models, such as GAE [Kipf and Welling, 2016a]. The prediction target is the existence of edge between a pair of nodes.

- *Attribute Masking* (denoted by AttrMasking). It is proposed by Hu et al. [2020a]. It masks node/edge attributes and then let GNNs predict those attributes based on neighboring structure.

- *Context Prediction* (denoted by ContextPred). It is also proposed by Hu et al. [2020a]. Context prediction uses subgraphs to predict their surrounding graph structures, and aims to mapping nodes appearing in similar structural contexts to nearby embeddings.

- *Graph Contrastive Learning* (denoted by GCL). It embeds augmented versions of the anchor close to each other (positive samples) and pushes the embeddings of other samples (negatives) apart. We use the augmentation strategies proposed in You et al. [2020] for generating the positive and negative samples.

To pre-train our models, we follow the training steps outlined in Hu et al. [2020a] for Infomax, EdgePred, AttrMasking, and ContextPred tasks. We then perform supervised graph-level property prediction to enhance the performance of the pre-trained models further. For models pre-trained using GCL, we follow the training steps detailed in You et al. [2020].

## B.3 Results of few-shot graph classification

**The results for 50-shot scenarios.** Table 4 summarizes the results for 50-shot graph classification.

Table 4: 50-shot test ROC-AUC (%) performance on molecular prediction benchmarks and protein function prediction benchmarks.

| Pre-training Strategy | Tuning Strategy | BBBP | Tox21 | ToxCast | SIDER | ClinTox | MUV | HIV | BACE | PPI | **Avg.** |
|---|---|---|---|---|---|---|---|---|---|---|---|
| Infomax | FT | 53.81 ±3.35 | 61.42 ±1.19 | 53.93 ±0.59 | 50.77 ±2.27 | 58.6 ±3.48 | 66.12 ±0.63 | 65.09 ±1.17 | 52.64 ±2.64 | 48.79 ±1.32 | 56.79 |
| | GPF | 55.52 ±1.84 | 65.56 ±0.64 | 56.76 ±0.54 | 50.29 ±1.61 | 62.44 ±4.11 | **68.00** ±0.61 | **67.68** ±1.09 | 54.49 ±2.54 | **54.03** ±0.34 | 59.41 |
| | GPF-plus | **58.09** ±2.12 | **65.71** ±0.37 | **57.13** ±0.48 | **51.33** ±1.14 | **62.96** ±3.27 | 67.88 ±0.42 | 66.80 ±1.43 | **56.56** ±6.81 | 53.78 ±0.45 | **60.02** |
| EdgePred | FT | 48.88 ±0.68 | 60.95 ±1.46 | 55.73 ±0.43 | 51.30 ±2.21 | 57.78 ±4.03 | 66.88 ±0.53 | 64.22 ±1.57 | 61.27 ±6.10 | 47.62 ±1.50 | 57.18 |
| | GPF | 50.53 ±1.35 | 64.46 ±0.93 | 57.33 ±0.65 | **51.35** ±0.76 | **68.74** ±6.03 | 68.08 ±0.39 | **66.22** ±1.90 | **62.85** ±5.91 | 52.81 ±0.38 | **60.26** |
| | GPF-plus | **54.49** ±4.60 | **64.99** ±0.53 | **57.69** ±0.61 | 51.30 ±1.18 | 66.64 ±2.40 | **68.16** ±0.48 | 62.05 ±3.39 | 62.60 ±2.48 | **53.30** ±0.34 | 60.13 |
| AttrMasking | FT | 51.26 ±2.33 | 60.28 ±1.73 | 53.47 ±0.46 | 50.11 ±1.63 | 61.51 ±1.45 | 59.35 ±1.31 | 67.18 ±1.59 | 55.62 ±5.04 | 48.17 ±2.45 | 56.32 |
| | GPF | 54.24 ±0.74 | 64.24 ±0.40 | **56.84** ±0.28 | 50.62 ±0.88 | 65.34 ±1.93 | 61.34 ±0.60 | 67.94 ±0.48 | **57.31** ±6.71 | 51.26 ±0.32 | 58.79 |
| | GPF-plus | **58.10** ±1.92 | **64.39** ±0.30 | 56.78 ±0.25 | 50.30 ±0.78 | 63.34 ±0.85 | **63.84** ±1.13 | **68.05** ±0.97 | 57.29 ±4.46 | **51.35** ±0.32 | **59.27** |
| ContextPred | FT | 49.45 ±5.74 | 58.77 ±0.70 | 54.46 ±0.54 | 49.89 ±1.16 | 48.60 ±3.40 | 56.14 ±1.84 | **60.91** ±1.90 | 56.37 ±1.90 | 46.33 ±1.76 | 53.43 |
| | GPF | 52.55 ±1.24 | 59.73 ±0.86 | 55.70 ±0.29 | 50.54 ±0.91 | **53.03** ±5.98 | 61.93 ±5.84 | 60.59 ±1.28 | 59.91 ±6.31 | 50.14 ±0.33 | 56.01 |
| | GPF-plus | **53.76** ±4.47 | **60.59** ±0.51 | **55.91** ±0.22 | **51.44** ±1.70 | 52.37 ±4.30 | **64.51** ±4.48 | 60.84 ±1.11 | **64.21** ±7.30 | **50.52** ±0.41 | **57.12** |
| GCL | FT | 54.40 ±2.87 | 48.35 ±1.67 | 50.29 ±0.19 | 53.23 ±0.87 | 54.05 ±4.16 | 46.73 ±1.88 | 60.05 ±3.80 | 49.87 ±1.78 | 49.94 ±1.77 | 51.62 |
| | GPF | 53.87 ±2.17 | **50.58** ±0.49 | 52.64 ±0.50 | 53.86 ±0.45 | 64.44 ±4.64 | 47.22 ±3.55 | **64.86** ±1.29 | **67.56** ±2.29 | 50.40 ±1.17 | 55.62 |
| | GPF-plus | **55.89** ±1.58 | 50.14 ±1.09 | **53.25** ±0.95 | **55.46** ±0.96 | **65.22** ±4.51 | **47.88** ±1.77 | 63.99 ±1.60 | 64.10 ±1.85 | **51.19** ±1.53 | 55.89 |

**The results for 100-shot scenarios.** We also conducted experiments where the number of training samples in the downstream task was limited to 100 for both the chemistry and biology datasets. The summary of the overall results can be found in Table 5. The experimental findings align with those observed in the 50-shot scenarios. Our proposed graph prompt tuning method achieves the best results in 42 out of 45 cases (14 out of 45 for GPF and 28 out of 45 for GPF-plus). Additionally, the average results of both GPF and GPF-plus surpass the average results of fine-tuning across all pre-training strategies, showcasing the superiority of our proposed graph prompt tuning approach.

## B.4 Parameter efficiency analysis

We have computed the number of tunable parameters for full fine-tuning, GPF, and GPF-plus (excluding the task-specific projection head $\theta$) on the chemistry and biology datasets. The statistics are presented in Table 6. The results indicate that the number of tunable parameters in GPF and GPF-plus is several orders of magnitude smaller than that of fine-tuning. Specifically, GPF utilizes no more than $0.02\%$ of the tunable parameters used in fine-tuning, while GPF-plus utilizes no more than $0.7\%$ of the tunable parameters used in fine-tuning. Our proposed graph prompt tuning methods

Table 5: 100-shot test ROC-AUC (%) performance on molecular prediction benchmarks and protein function prediction benchmarks.

| Pre-training Strategy | Tuning Strategy | BBBP | Tox21 | ToxCast | SIDER | ClinTox | MUV | HIV | BACE | PPI | Avg. |
|---|---|---|---|---|---|---|---|---|---|---|---|
| Infomax | FT | 56.29 ±1.65 | 60.46 ±0.66 | 55.34 ±0.20 | 50.49 ±1.29 | 50.90 ±4.92 | 65.88 ±1.76 | 65.81 ±1.43 | 57.35 ±2.67 | 49.74 ±0.72 | 56.91 |
| | GPF | 56.38 ±2.84 | 61.54 ±0.28 | 57.31 ±0.35 | 54.49 ±0.72 | 56.49 ±1.98 | 66.52 ±0.67 | **68.02** ±1.22 | 61.67 ±2.76 | 54.57 ±0.51 | 59.66 |
| | GPF-plus | **56.97** ±3.46 | **62.48** ±0.80 | **57.64** ±0.30 | **54.86** ±0.49 | **57.68** ±0.89 | 67.00 ±0.47 | 67.66 ±1.13 | **61.76** ±2.80 | **54.66** ±0.52 | **60.07** |
| EdgePred | FT | 51.27 ±3.89 | 61.48 ±1.21 | 58.28 ±0.81 | 52.23 ±1.27 | 58.50 ±2.54 | 64.32 ±2.48 | 59.82 ±1.47 | **50.86** ±0.85 | 48.06 ±2.00 | 56.09 |
| | GPF | **55.13** ±1.27 | 63.35 ±0.94 | 59.09 ±0.55 | 52.30 ±0.54 | **65.02** ±4.13 | 65.47 ±0.31 | **63.19** ±1.49 | 48.64 ±1.70 | 52.52 ±0.46 | 58.30 |
| | GPF-plus | 54.20 ±5.05 | **64.80** ±0.95 | **59.42** ±0.23 | **52.47** ±0.64 | 62.73 ±3.54 | 65.37 ±0.37 | 63.18 ±1.28 | 50.02 ±5.65 | **53.00** ±0.44 | **58.35** |
| AttrMasking | FT | 54.56 ±4.82 | 60.95 ±1.28 | 55.84 ±0.40 | 50.64 ±1.16 | 61.16 ±1.19 | 64.90 ±1.43 | 61.65 ±3.31 | 59.03 ±2.89 | 47.29 ±1.43 | 57.33 |
| | GPF | **55.23** ±3.14 | 63.36 ±0.61 | 57.66 ±0.34 | 50.08 ±0.59 | **63.05** ±4.41 | 65.58 ±0.69 | **69.79** ±1.78 | **59.37** ±3.90 | **52.31** ±0.41 | **59.60** |
| | GPF-plus | 53.58 ±2.19 | **63.89** ±0.58 | **57.72** ±0.37 | **51.70** ±0.61 | 62.68 ±2.50 | **66.47** ±0.43 | 69.35 ±1.58 | 58.50 ±2.36 | 52.28 ±0.89 | 59.57 |
| ContextPred | FT | 50.42 ±0.57 | 60.74 ±0.88 | 56.00 ±0.29 | 51.81 ±1.77 | 51.48 ±2.86 | 64.87 ±2.30 | 59.82 ±2.00 | 50.43 ±3.74 | 45.39 ±0.42 | 54.55 |
| | GPF | 52.33 ±5.07 | 63.91 ±0.82 | 57.32 ±0.30 | 53.55 ±0.88 | **54.31** ±2.58 | 65.80 ±0.45 | 68.51 ±2.23 | **54.70** ±5.89 | 50.44 ±0.64 | 57.87 |
| | GPF-plus | **53.62** ±6.59 | **64.89** ±0.89 | **58.02** ±0.52 | **54.13** ±1.38 | 54.02 ±2.38 | **65.89** ±0.54 | **68.75** ±3.80 | 54.41 ±5.85 | **50.79** ±0.50 | **58.28** |
| GCL | FT | 44.06 ±2.55 | 48.47 ±1.63 | 51.91 ±0.33 | **56.10** ±0.45 | 48.13 ±3.23 | 53.93 ±0.91 | 32.63 ±0.93 | **55.41** ±1.28 | 49.44 ±1.53 | 48.89 |
| | GPF | 51.34 ±1.01 | 55.46 ±1.54 | **53.78** ±0.58 | 53.37 ±0.99 | **60.44** ±4.10 | 54.06 ±2.56 | 44.23 ±0.75 | 49.20 ±2.94 | 54.35 ±0.65 | 52.91 |
| | GPF-plus | **52.47** ±1.19 | **57.42** ±1.36 | 53.07 ±0.81 | 52.90 ±1.05 | 60.22 ±3.81 | **55.68** ±1.25 | **46.24** ±3.35 | 51.64 ±4.25 | **54.47** ±1.25 | **53.79** |

exhibit significant advantages in terms of parameter efficiency compared to fine-tuning. It leads to reduced training time and storage space required for downstream adaptations.

Table 6: The number of tunable parameters for different tuning strategies.

| Dataset | Tuning Strategy | Tunable Parameters | Relative Ratio (%) |
|---|---|---|---|
| Chemistry | FT | $\sim 1.8M$ | 100 |
| | GPF | $\sim 0.3K$ | 0.02 |
| | GPF-plus | $\sim 3\text{-}12K$ | 0.17-0.68 |
| Biology | FT | $\sim 2.7M$ | 100 |
| | GPF | $\sim 0.3K$ | 0.01 |
| | GPF-plus | $\sim 3\text{-}12K$ | 0.11-0.44 |

## B.5 Comparison with linear probing

In the field of Natural Language Processing and Computer Vision, *linear probing* [Kumar et al., 2022, Wu et al., 2020, Tripuraneni et al., 2020, Du et al., 2020] is a widely employed method for adapting pre-trained models to downstream tasks. This approach involves freezing the parameters of the pre-trained model $f$ and solely optimizing the linear projection head $\theta$. To evaluate the effectiveness of linear probing, we conducted experiments on the chemistry datasets Toxcast and SIDER, and the results are summarized in Table 7. It is evident from the results that linear probing exhibits a significant performance decline compared to fine-tuning and our proposed graph prompt tuning. The primary distinction between linear probing and our proposed graph prompt tuning lies in the incorporation of an additional learnable graph prompt $g_\phi(\cdot)$ in the input space. The substantial

Table 7: Test ROC-AUC (%) performance on Toxcast and SIDER with the linear probing.

| Pre-training Strategy | Tuning Strategy | ToxCast | SIDER | Avg. |
|---|---|---|---|---|
| Infomax | FT | 65.16 ±0.53 | 63.34 ±0.45 | 64.25 |
| | Linear Probing | 63.84 ±0.10 | 59.62 ±0.73 | 61.73 |
| | GPF | 66.10 ±0.53 | **66.17** ±0.81 | **66.13** |
| | GPF-plus | **66.35** ±0.37 | 65.62 ±0.74 | 65.98 |
| EdgePred | FT | **66.29** ±0.45 | 64.35 ±0.78 | 65.32 |
| | Linear Probing | 65.25 ±0.09 | 61.47 ±0.03 | 63.36 |
| | GPF | 65.65 ±0.30 | 67.20 ±0.99 | 66.42 |
| | GPF-plus | 65.94 ±0.31 | **67.51** ±0.59 | **66.72** |
| AttrMasking | FT | 65.34 ±0.30 | 66.77 ±0.13 | 66.05 |
| | Linear Probing | 64.75 ±0.07 | 62.60 ±0.57 | 63.67 |
| | GPF | 66.32 ±0.42 | **69.13** ±1.16 | **67.72** |
| | GPF-plus | **66.58** ±0.13 | 68.65 ±0.72 | 67.61 |
| ContextPred | FT | 66.39 ±0.57 | 64.45 ±0.60 | 65.42 |
| | Linear Probing | 65.35 ±0.09 | 61.28 ±0.39 | 63.31 |
| | GPF | **67.92** ±0.35 | 66.18 ±0.46 | 67.05 |
| | GPF-plus | 67.58 ±0.54 | **66.94** ±0.95 | **67.26** |
| GCL | FT | 62.54 ±0.26 | 60.63 ±1.26 | 61.58 |
| | Linear Probing | 50.92 ±0.22 | 52.91 ±0.62 | 51.91 |
| | GPF | 62.70 ±0.46 | 61.26 ±0.53 | 61.98 |
| | GPF-plus | **62.76** ±0.75 | **62.37** ±0.38 | **62.56** |

performance gap observed between these approaches underscores the importance of integrating a graph prompt for the effective adaptation of pre-trained models.

## B.6 Comparison with other tuning methods

We also compare our proposed graph prompt tuning with other tuning methods described as follows:

- PARTIAL-$k$: We tune the last $k$ layers of the pre-trained model $f$ with a projection head $\theta$ and freeze other parts, which is utilized in Zhang et al. [2016], He et al. [2021], Jia et al. [2022].
- MLP-$k$: We freeze the pre-trained model $f$ and utilize a multi-layer perceptron (MLP) with $k$ layers as the projection head to perform the classification.

We conduct the experiments on the biology datasets (PPI), and Table 8 summarizes the results. The experimental results indicate that our methods outperform other tuning methods in all cases.

## B.7 Extra results on GCC

Another popular pre-training strategy for graph contrastive learning involves following the training steps outlined in Qiu et al. [2020a]. First, we introduce the datasets utilized for pre-training GCC. The self-supervised pre-training task of GCC is conducted on six graph datasets, and Table 9 provides detailed statistics for each dataset. For the downstream tasks of the pre-trained GCC, we employ

Table 8: Test ROC-AUC (%) performance on protein function prediction benchmarks with different tuning methods.

| Pre-training Strategy | FT | MLP-3 | Partial-1 | Partial-3 | GPF | GPF-plus |
|---|---|---|---|---|---|---|
| Infomax | 71.29 ±1.79 | 74.68 ±0.56 | 74.36 ±0.92 | 73.28 ±0.18 | 77.02 ±0.42 | **77.03** ±0.32 |
| EdgePred | 71.54 ±0.85 | 74.60 ±0.88 | 73.24 ±0.68 | 73.35 ±0.77 | 76.98 ±0.20 | **77.00** ±0.12 |
| AttrMasking | 73.93 ±1.17 | 77.99 ±0.42 | 75.91 ±0.10 | 74.02 ±0.37 | **78.91** ±0.25 | 78.90 ±0.11 |
| ContextPred | 72.10 ±1.94 | 76.01 ±0.68 | 76.62 ±0.92 | 74.86 ±0.79 | 77.42 ±0.07 | **77.71** ±0.21 |

IMDB-BINARY and IMDB-MULTI datasets [Yanardag and Vishwanathan, 2015]. Each dataset consists of a collection of graphs associated with specific target labels. We evaluate GPF on these datasets, and the results are presented in Table 10. The experimental findings consistently demonstrate that GPF outperforms fine-tuning when adapting models pre-trained using the GCC strategy.

Table 9: Statistics of datasets for pre-training

| Dataset | Academia | DBLP(SNALP) | DBLP(NetRep) | IMDB | Facebook | LiveJournal |
|---|---|---|---|---|---|---|
| $|V|$ | 137,969 | 317,080 | 540,486 | 896,305 | 3,097,165 | 4,843,953 |
| $|E|$ | 739,984 | 2,099,732 | 30,491,158 | 7,564,894 | 47,334,788 | 85,691,368 |

Table 10: Test accuracy (%) performance of GCC on graph classification benchmarks.

| Pre-training Strategy | Tuning Strategy | IMDB-B | IMDB-M | Avg. |
|---|---|---|---|---|
| GCC (E2E) | FT | 72.60 ±4.72 | 49.07 ±3.59 | 60.83 |
| | GPF | **73.40** ±3.80 | **49.17** ±3.12 | **61.28** |
| GCC (MoCo) | FT | 71.70 ±4.98 | 48.07 ±2.91 | 59.88 |
| | GPF | **72.50** ±3.20 | **49.33** ±3.93 | **60.91** |

## B.8   Hyper-parameter settings

This section presents the hyper-parameters used during the adaptation stage of pre-trained GNN models on downstream tasks for our proposed graph prompt tuning. Table 11 summarizes the hyper-parameter settings. You can also visit our code repository to obtain the specific commands for reproducing the experimental results.

Table 11: The hyper-parameter settings.

| Dataset | Pre-training Strategy | Prompt Dimension | Learning Rate | Weight Decay | Batch Size | Training Epoch |
|---|---|---|---|---|---|---|
| Biology | Infomax | 300 | 0.001 | 0 | 32 | 50 |
| | EdgePred | 300 | 0.001 | 0 | 32 | 50 |
| | Masking | 300 | 0.001 | 0 | 32 | 50 |
| | ContextPred | 300 | 0.001 | 0 | 32 | 50 |
| | GCL | 300 | 0.0001 | 0 | 32 | 50 |
| Chemistry | Infomax | 300 | 0.001 | 0 | 32 | 100 |
| | EdgePred | 300 | 0.001 | 0 | 32 | 100 |
| | Masking | 300 | 0.001 | 0 | 32 | 100 |
| | ContextPred | 300 | 0.001 | 0 | 32 | 100 |
| | GCL | 300 | 0.001 | 0 | 32 | 100 |
| IMDB-B | GCC | 64 | 0.005 | Linear | 128 | 100 |
| IMDB-M | GCC | 64 | 0.005 | Linear | 128 | 100 |

