# OpenReview forum: "Universal Prompt Tuning for Graph Neural Networks"
_NeurIPS.cc/2023/Conference — NeurIPS 2023 poster_

### Official Review · Reviewer_KwHM · 2023-07-03

**Soundness:** 2 fair
**Presentation:** 3 good
**Contribution:** 2 fair
**Rating:** 5
**Confidence:** 4

**Summary:**

The paper introduces a universal prompt-based tuning method called Graph Prompt Feature (GPF) and its variation (GPF-plus) for pre-trained Graph Neural Network (GNN) models. GPF is a universal method that can be applied to any pre-trained GNN model under any pre-training strategy. It operates on the input graph's feature space and can achieve an equivalent effect to any form of prompting function.
GPF introduces a learnable vector $p$ of dimension $F$, which is added to the node features, where $F$ corresponds to the dimensionality of the node features. The authors also purpose GPF-plus, which assigns an independent learnable vector $p_i$ to each node $u_i$ in the graph, instead of a single vector $p$. Experimental results show that GPF outperforms fine-tuning, with an average improvement of about 1.4% in full-shot scenarios and 3.2% in few-shot scenarios.

**Strengths:**

1) Theoretical Guarantees: The authors provide rigorous derivations and theoretical analyses to demonstrate the universality and effectiveness of GPF. They prove that GPF can achieve an equivalent effect to any form of prompting function and can outperform fine-tuning in certain cases. This theoretical foundation strengthens the credibility of their proposed method.

2) Experimental Validation - Reproducibility: The paper includes extensive experiments conducted under various pre-training strategies, including both full-shot and few-shot scenarios. The experimental results consistently show that GPF outperforms fine-tuning, achieving an average improvement of about 1.4% in full-shot scenarios and 3.2% in few-shot scenarios. Furthermore, GPF surpasses existing specialized prompt-based tuning methods designed for specific pre-training strategies. Moreover, the availability of the source code enhances the reproducibility of the experiments.



**Weaknesses:**

1) A weakness of the paper is the lack of a clear explanation or motivation regarding why the addition of the learnable vector p to the input features in the GPF model leads to better results compared to linear probing with a trainable layer in the final layer. While the Appendix demonstrates the superior performance of GPF, the paper does not provide a comprehensive analysis or reasoning behind this improvement. The absence of a clear explanation may leave readers questioning the underlying mechanisms and factors that contribute to the observed performance gain. Without a proper understanding of the motivations and justifications for the proposed approach, it becomes challenging to assess the significance and generalizability of the findings.

2) Theorem 2 assumes a simple linear 1-layer GNN (without activations). While the theorem provides theoretical insights into the convergence properties of this specific type of GNN, it may not accurately capture the behavior of more complex GNN architectures commonly used in practice. In real-world applications, GNNs often incorporate activation functions to introduce non-linearity and capture more intricate patterns in graph data. By focusing solely on a linear 1-layer GNN without activations, the theorem may limit the generalizability of its conclusions and overlook important aspects of GNN models commonly used in practical scenarios. Therefore, the applicability of the theorem's findings to more sophisticated GNN architectures with non-linear activations remains uncertain.

3) One weakness of the paper is the exclusive use of Graph Isomorphism Network (GIN) as the backbone GNN for finetuning. While GIN is a widely used GNN architecture, it may not be the most expressive or optimal choice for every graph-related task[1,2]. The paper does not provide a comprehensive exploration or comparison of different backbone GNN architectures in the finetuning process. Moreover, maybe a more powerful GNN would be able to achieve better performance in the fine-tuning stage. Therefore, it would be interesting for the authors to examine if their approach leads also to better results when a more powerful GNN is used.

[1] Frasca, Fabrizio, et al. "Understanding and extending subgraph gnns by rethinking their symmetries." Advances in Neural Information Processing Systems 35 (2022): 31376-31390.
[2]  Morris, Christopher, et al. "Weisfeiler and leman go neural: Higher-order graph neural networks." Proceedings of the AAAI conference on artificial intelligence. Vol. 33. No. 01. 2019.





**Questions:**

1) Explanation of the relationship between GPF and linear probing: The paper compares GPF with linear probing, which involves a trainable layer in the final layer. It would be beneficial if the authors could discuss the relationship between these two approaches and explain why GPF, with its learnable vector added to the input features, achieves superior performance compared to linear probing. Is there any inherent advantage or characteristic of GPF that enables it to outperform linear probing?

2) Generalizability of Theorem 2: The paper presents Theorem 2, which focuses on a simple linear 1-layer GNN without activations. However, many practical GNN architectures incorporate activation functions to introduce non-linearity. It would be helpful if the authors could discuss the generalizability of the theorem's conclusions to more complex GNN models commonly used in real-world applications. Can the findings of Theorem 2 be extended to GNNs with non-linear activations? If not, what are the limitations or implications of the theorem in the context of practical GNN architectures?

3) Have you considered or experimented with using more powerful GNN architectures, such as subgraph GNNs or higher order GNNs, as the backbone GNN for the finetuning stage? If so, how does the performance of GPF and GPF-plus compare when utilizing these alternative GNN architectures?

I will be happy to increase my score for the paper, if the authors adequately address the mentioned weaknesses and provide satisfactory explanations and improvements in response to the questions and suggestions raised.


*** After rebuttal increased score to "Borderline Accept" **

**Limitations:**

yes

---

> ### Author Rebuttal · Authors · 2023-08-09
>
> ### **Response**
>
> Dear reviewer KwHM,
> We hope our point-to-point responses can address your concerns and provide you with better clarification.
>
> 1. (Weakness 1 \& Question 1) Comparison with linear probing.
>
> The difference between linear probing and GPF lies in the introducing of additional learnable parameters in GPF that modify the input graph. While linear probing focuses solely on training the last linear head $\theta$ of the model, GPF not only trains $\theta$ but also introduces a learnable vector $p$ into the input graph's feature space. To emphasize the theoretical advantage of GPF over linear probing, we define the pre-trained model as $f$, the input graph as $G$, the linear head as $\theta$, and the target matrix as $Y$. Let $f(G)=H$ represent the representation matrix obtained from the pre-trained model. In linear probing, the goal is to find the optimal $\theta$ that minimizes the discrepancy between $H\theta$ and $Y$. Meanwhile, GPF incorporates additional learnable parameters $p$ into the input graph $G$, resulting in $f(G+p)=H^\prime$. When applying GPF, we can simultaneously adjust $H^\prime$ and $\theta$ to make $H^\prime\theta$ close to $Y$. This simultaneous adjustment yields better results compared to only modifying $\theta$, thus establishing the theoretical superiority of GPF over linear probing. Notably, linear probing cannot modify the representations $H$ obtained from the pre-trained model. In contrast, GPF addresses this limitation by manipulating the input to modify the representations obtained by the pre-trained model. Additionally, the comparison between GPF and linear probing can be regarded as an extension of Theorem 2. When the GNN model does not contain any learnable parameters, Theorem 2 highlights the theoretical advantage of GPF over linear probing. Furthermore, akin to the success of prompt tuning in the NLP domain, the merit of GPF can also be attributed to its capacity to bridge the disparity between pre-training and downstream tasks through input transformation. Consequently, compared to linear probing, GPF exhibits advantages in both intuition and theory.
>
> 2. (Weakness 2 \& Question 2) Generalizability of Theorem 2.
>
> Theorem 2 can also be applied to multi-layer GNN models, albeit with slight variations in some coefficients during the derivation process. However, incorporating non-linear activation functions presents a significant challenge for theoretical analysis. The crux lies in accurately estimating the expressive power of the multi-layer perceptron (MLP) architecture within the model. The universal approximation theorem describes the upper bound on the expressive power of MLP, stating that MLP networks with infinite depth or infinite width can theoretically approximate any function. Consequently, comparing the theoretical optimal expressive power of any network with an MLP architecture becomes meaningless. However, in practical scenarios, especially for GNN models with limited dimensions and depth, the real impact of multi-layer linear transformations with non-linear activation functions is not significantly different from that of single-layer linear transformations [1,2]. Therefore, similar to many existing works that analyze the theoretical capacity of models [3,4], we omit the consideration of non-linear activation functions in Theorem 2. Nonetheless, the presented derivation can be generalized to practical applications with only minor discrepancies.
>
> 3. (Weakness 3 \& Question 3) Results on more powerful GNN backbones.
>
> Your suggestion is rather rational, and we have included additional experimental results using more powerful GNN backbones. The results can be found in B.1 of the global response. From the experimental results, we can find that our method still achieves better results on these powerful models.
>
> We hope our explanation can dispel your concerns. If you have any other questions or concerns, please feel free to let us know.
>
>
>
> **Ref**
>
> [1] Simplifying Graph Convolutional Networks
>
> [2] DFG-NAS: Deep and Flexible Graph Neural Architecture Search
>
> [3] Fine-Tuning can Distort Pretrained Features and Underperform Out-of-Distribution
>
> [4] In Search of the Real Inductive Bias: On the Role of Implicit Regularization in Deep Learning

---

> > ### Comment · Reviewer_KwHM · 2023-08-16
> > **Response to Authors**
> >
> > I would like to thank the authors for their rebuttal and the new experimental results. I increase my score to "borderline accept".

---

> > > ### Author Response · Authors · 2023-08-20
> > >
> > > Thanks for your support in our work. Your valuable feedback has made our work better.

---

### Official Review · Reviewer_dbJu · 2023-07-04

**Soundness:** 3 good
**Presentation:** 3 good
**Contribution:** 3 good
**Rating:** 6
**Confidence:** 4

**Summary:**

This paper proposed a universal prompt-based tuning method, called GPF, for pre-trained GNN models. The idea is to operate on the feature space of the downstream input graph. The authors theoretically showed that GPF can achieve results equivalent to any prompting function, and is not weaker than full fine-tuning. Experiments showed the competitive results.

**Strengths:**

1. This paper proposed a universal graph prompting approach, which basically operates on the input graph feature space. As far as I know, the idea of augmenting input graph node features for prompting is novel.

2. The authors provide theoretical analysis, which showed that GPF can achieve results equivalent to any prompting function, and is not weaker than full fine-tuning.

3. Experiments results showed the performance gains.

**Weaknesses:**

1. The motivation for a universal prompting needs better explained, and advantages over specialized prompting approaches.

2. Comparison with other prompting meothds is only perfomed on the models pre-trained by edge predcition.

Some minor comments:
1. It is not clear to me how the avg. in Table 1 and Table 2 obtained, and what does it mean. It is just the average across all datasets?
2. I suggest the authors include the detailed pretraining settings in experiments (e.g., datasets used) instead of in the appendix.
3. Fig. 1 is not informative. The authors should use a figure to better illustrate how GPF works, and the main differences with existing prompting approaches.
4. Table captions should be on top of tables insead of below.

**Questions:**

See the weaknesses.



**Limitations:**

Potential limitations of the proposed GPF not discussed.

---

> ### Author Rebuttal · Authors · 2023-08-09
>
> ### **Response**
>
> Dear reviewer dbJu,
> We really appreciate your comments on our work. We hope our response can address your concerns.
>
> 1. (Weakness 1) The motivation and advantages of universal prompting.
>
> The universal graph prompt tuning method that we proposed offers three main advantages over specialized prompting methods:
>
> a. Generalization without knowledge of pre-training details: One of the primary advantages of our method is that it eliminates the need for an in-depth understanding of the pre-training strategies employed by the pre-trained GNN models. In practical applications, gathering such detailed knowledge can be challenging and inconvenient. Our universal method can achieve satisfactory results without requiring extra information about the model's pre-training details.
>
> b. Applicability to complex graph pre-training tasks: Existing graph prompt tuning methods predominantly focus on the pre-training task of link prediction. However, when dealing with more abstract and complex graph pre-training tasks, such as graph infomax and graph contrastive learning, it becomes challenging for researchers to design appropriate prompt templates intuitively. Our universal method circumvents the need for an explicit characterization of concrete prompt templates, thus making it applicable to models trained on any graph pre-training task.
>
> c. Practical and theoretical effectiveness: Our universal method is easy to apply in practice and comes with strict theoretical guarantees of effectiveness. It offers valuable insights for future investigations in this field.
>
> We appreciate your suggestion to include additional discussion, and we have added similar content in our latest revision.
>
>
>
> 2. (Weakness 2) Comparing existing graph prompt tuning methods under alternative pre-training strategies.
>
> We compare our approach with existing graph prompt tuning methods in the context of edge prediction because both GPPT and GraphPrompt explicitly state in their papers that they are designed specifically for the pre-training task of link prediction. These methods explicitly transform node classification tasks into link prediction tasks, which limits their ability to bridge the gap between upstream and downstream tasks when applied to other pre-training strategies. To provide more evidence, we present additional experimental results that indicate the limitations of these methods in dealing with other pre-training tasks. As an example, the table below shows the ROC-AUC scores (%) of these methods on the model pre-trained by graph infomax.
>
> |         |             | BBBP           | Tox21         | ToxCast        | SIDER          | ClinTox        | BACE           |
> | ------- | ----------- | -------------- | ------------- | -------------- | -------------- | -------------- | -------------- |
> | Infomax | FT          | **67.55±2.06** | 78.57±0.51    | 65.16±0.53     | 63.34±0.45     | 70.06±1.45     | 81.32±1.25     |
> |         | GPPT        | 56.92±0.46     | 59.37±0.84    | 53.73±0.92     | 48.23±0.79     | 53.22±0.84     | 56.22±0.16     |
> |         | GPPT(w/olo) | 62.87±0.05     | 71.50±0.70     | 57.55±0.13     | 55.77±0.19     | 54.49±0.40     | 75.37±0.30     |
> |         | GraphPrompt | 62.95±0.89     | 61.65±0.40    | 54.98±0.32     | 51.21±0.21     | 47.53±0.12     | 54.77±0.99     |
> |         | GPF         | 66.83±0.86     | 79.09±0.25    | 66.1±0.53      | **66.17±0.81** | 73.56±3.94     | 83.60±1.00     |
> |         | GPF-plus    | 67.17±0.36     | **79.13±0.70** | **66.35±0.37** | 65.62±0.74     | **75.12±2.45** | **83.67±1.08** |
>
>
>
> 3. Collection of minor comments.
>
> (a) You are right, and the Avg. in Table 1 and 2 presents the average values across all datasets.
>
> (b) Thanks for your suggestion. We have moved the detailed experiment settings in our main text.
>
> (c) Thanks for your suggestion. We have adjusted the last two figures to make their meaning clearer.
>
> (d) Thanks for your detailed advice, and we have resolved this mistake.

---

> > ### Comment · Reviewer_dbJu · 2023-08-16
> >
> > Thanks for the rebuttal. I have read comments from other reviewers, as well as rebuttals, and I think my score is resonable and will keep my rating.

---

> > > ### Author Response · Authors · 2023-08-20
> > >
> > > Thanks for your support in our work. Your valuable feedback has made our work better.

---

### Official Review · Reviewer_Cfab · 2023-07-06

**Soundness:** 2 fair
**Presentation:** 3 good
**Contribution:** 2 fair
**Rating:** 5
**Confidence:** 4

**Summary:**

This paper aims for efficient adaptation of pre-trained graph neural networks to downstream tasks. A simple prompt tuning method (i.e. GPF) is proposed for adaptation and is applicable to GNN pretrained with any objectives. The main idea of GPF is to add a learnable vector on all the node features in the input graph, while an improved version (i.e. GPF-plus) leverages a set of learnable vectors to compose the bias added to each node feature. Experiments show that GPF and GPF-plus achieve better results than the full finetuning.

**Strengths:**

Strength
1.	Figure 1 illustrates the difference between the proposed method and existing methods.

2.	Provide a theoretical analysis of the proposed method.

3.	The proposed method is simple and easy to understand.

**Weaknesses:**

Major
1.	The novelty of GSF may be limited because it just adds a bias term on the input of the pretrained network and there is nothing specific for the graph structure. However, the bias-term finetuning is not a new idea [a, b].

2.	The authors claim that the developed method is universal for all the graph learning. However, the theoretical proof is only valid for graph classification downstream tasks where the node features are sum-pooled as the representation. There is no evidence showing that GSF is applicable to other types of downstream tasks.

3.	Baselines missing in Table 1 & 2. For example, Linear Probing is a standard baseline, and residual adapter in [c] can also be a baseline.

4.	Given the variance shown in Table 1 & 2, the improvement of GPF-plus over GPF is marginal. Why is it an enhanced version? If GPF is theoretically feasible to address the problem (section 3.4), why there GPF-plus is needed?

[a] BitFit: Simple parameter-efficient fine-tuning for transformer-based masked language-models. ACL’22.
[b] Tinytl: Reduce memory, not parameters for efficient on-device learning. NeurIPS’20.
[c] CLIP-Adapter: Better Vision-Language Models with Feature Adapters.

**Questions:**

The author is suggested to address the concerns in the weakness section.

**Limitations:**

Limitation is not discussed in the paper.

---

> ### Author Rebuttal · Authors · 2023-08-09
>
> ### **Response**
>
> Dear reviewer Cfab,
> We hope our point-to-point responses can address your concerns and better clarify the contributions and value of our work.
>
> 1. (Weakness 1) The novelty of GPF and the comparison with existing methods.
>
> The main contributions of our work compared to existing methods can be summarized as follows:
>
> a. We have pioneered the general prompt tuning method on pre-trained GNN models, offering rigorous derivations to fill the gap in the lack of theoretical validity for the effectiveness of graph prompt tuning.
> Specifically, GPF can be seen as a general graph template theoretically applicable to any pre-training task. Despite its relatively simple form, GPF's effectiveness in bridging the gap between pre-training and downstream tasks is theoretically guaranteed (Theorem 1). While the bias-term tuning methods you mentioned are designed for parameter efficiency, our method stands out by considering the influence of the pre-training task, which distinguishes our method from conventional parameter tuning techniques.
>
> b. Our work provides valuable insights that can guide future investigations in this field. We aim to find a universal prompt tuning method that can provide effective prompts for any graph pre-training strategy. Our paper introduces GPF and GPF-plus as the most intuitive and concise options for universal graph prompt tuning. Based on our deduction and analysis, designing more complex prompt tuning methods is possible. We highlight an updated work, "All in One: Multi-task Prompting for Graph Neural Networks", which received the best research paper award of SIGKDD 2023 recently and successfully incorporated our innovative ideas with multi-task meta-learning. Our work serves as its theoretical foundation, providing indispensable theoretical guarantees for their proposed method.
>
> Our method, GPF, operates on the feature space of the input graph, but it does not imply that we overlook modifications to the graph structure. While we focus on adding extra learnable vectors to the feature space, it is important to note that we have demonstrated the ability of this operation to achieve an equivalent effect to any structure modification. Detailed proofs regarding this equivalence can be found in Propositions 4 and 5 in the appendix. Consequently, our method does not need explicit structural modifications. There are several advantages to this simplification. The adjacency matrix, an N*N matrix, is much larger than the feature matrix. By working on the feature matrix, our method significantly reduces resource consumption and minimizes training difficulties.
>
> 2. (Weakness 2) Applicability of theoretical proof and other downstream tasks.
>
> The theoretical proof provided can also be extended to node-level tasks. From a subgraph aggregation perspective, the node representations can be considered as the subgraph representations of the central nodes. This means that obtaining node representations can be transformed into obtaining subgraph representations. To further explain the application of GPF on node-level tasks, please refer to Section A.1 in the appendix. Furthermore, in Section 3.4 of the main text, we demonstrate that GPF achieves high flexibility and effectiveness in generating graph representations. This implies that when applied to node-level tasks, GPF can also obtain sufficiently flexible and effective node representations. We are sorry to cause your concern and want to assure you that we have included a theoretical discussion on the effectiveness of GPF for node-level tasks in our latest revision. To address your concerns, we also provide additional empirical results on node classification tasks, which can be found in B.1 of the global response.
>
> Regarding the sum pooling assumption in the theorem derivation, it is not a necessary condition for the theorem to hold. Our analysis can be extended to accommodate any weighted aggregation readout function, including average pooling, max/min pooling, and classical hierarchical pooling. These pooling methods are all encompassed within our analysis. We use the sum pooling assumption to facilitate a straightforward and comprehensible analysis. And we have included additional discussions on other pooling techniques in our latest revision.
>
> 3. (Weakness 3) The comparison with linear probing and residual adapter.
>
> We have already conducted a thorough performance comparison of our method with linear probing, and the corresponding results can be found in Section B.5 within the appendix. Additionally, we have evaluated our method against other classical tuning methods, and the corresponding results can be found in Section B.6 within the appendix. Regarding your suggestion to compare our method with the residual adapter, we have taken it into consideration and provided the results in B.3 of the global response.
>
>
> 4. (Weakness 4) Advantages of GPF-plus over GPF.
>
> We introduce GPF-plus to enhance the expressiveness and scalability of our method. Both GPF and GPF-plus are theoretically guaranteed to be universal. However, GPF-plus offers greater flexibility by allowing for different prompted features to be provided for each node.
> When GPF and GPF-plus are employed to approximate a specific graph template, GPF-plus possesses a larger set of feasible solutions than GPF due to its enhanced flexibility. Consequently, GPF-plus has a greater chance of obtaining better solutions during the practical training stage. Additionally, while the number of parameters in GPF is fixed according to the dimensionality of the node features, GPF-plus provides the option to adjust the number of parameters freely by selecting the number of prompt feature bases. This adaptability enables GPF-plus to flexibly adjust to the characteristics of downstream datasets.
>
> We hope our explanation can dispel your concerns. If you have any other questions or concerns, please feel free to let us know.

---

> > ### Comment · Reviewer_Cfab · 2023-08-18
> > **Thank you for rebuttal**
> >
> > Thanks the author for providing the rebuttal. My concerns have been addressed and the score will be increased.

---

> > > ### Author Response · Authors · 2023-08-20
> > >
> > > Thanks for your support in our work. Your valuable feedback has made our work better.

---

### Official Review · Reviewer_GUF8 · 2023-07-06

**Soundness:** 3 good
**Presentation:** 2 fair
**Contribution:** 2 fair
**Rating:** 5
**Confidence:** 4

**Summary:**

This paper introduces the Graph Prompt Feature (GPF) approach, which aims to adapt pre-trained Graph Neural Networks (GNNs) for downstream tasks by appending tunable embeddings onto the frozen node embeddings. By doing so, the authors achieve a significant reduction in the number of parameters required for the downstream task compared to full fine-tuning. The experimental results demonstrate promising performance in binary classification tasks on Molecular and Protein datasets when compared to full fine-tuning.

**Strengths:**

The paper is clearly written, the contribution is easy to understand and provide theoretical proof on effectiveness.

**Weaknesses:**

There are several concerns that I would like to see addressed:

1) In the formulation of GPF, its relation to the concept of prompting seems unclear. It is challenging to connect this approach to the motivating concept of prompting in NLP. Instead, GPF appears to resemble task-specific adaptive fine-tuning in transformers, such as LoRA[1]. It would be helpful if the authors could elaborate more on where the similarity between GPF and prompting lies.

2) The choice of downstream tasks seems to be limited to small binary classification tasks, and the comparison is only made against fine-tuning as a baseline. Why are there no results presented for multi-class classification tasks? This decision weakens the potential of the proposed method.

3) Unlike previous graph prompt methods, GPF does not make any modifications to the graph's structure. It would be beneficial if the authors could explain how graph classification highlights the strength of GPF, as the additional results provided may seem somewhat redundant.

4) There is no detail provided on the number of features appended into the embedding space for each node. Since GPF-plus suggests a simple size range, does this mean that the authors used different settings for GPF-plus for each dataset? This is not discussed in the experimental settings and might make the comparison somewhat incomparable.

[1] https://arxiv.org/abs/2106.09685

**Questions:**

see weaknesses

**Limitations:**

The author did not address any limitations.

---

> ### Author Rebuttal · Authors · 2023-08-09
>
> ### **Response**
>
> Dear reviewer GUF8,
> We hope our point-to-point responses can address your concerns and provide you with better clarification of the contributions and value of our work.
>
>
> 1. (Weakness 1) The relationship between our methods and prompting methods.
>
> Our method, GPF, is a general prompt tuning method that provides theoretically guaranteed prompts for any pre-training task within the context of GNNs. What distinguishes GPF from conventional parameter tuning techniques, such as LoRA, is its consideration of the influence of the pre-training task. Specifically, in the field of NLP, prompt tuning methods transform downstream tasks into sentence completion tasks to align them more closely with the pre-training tasks. Similarly, in the field of graphs, researchers aim to bridge the gap between downstream and pre-training tasks for GNN models. Graph prompt tuning involves selecting suitable graph templates based on the pre-training tasks, and this process is formally described in Section 3.2 of our paper. All existing graph prompt tuning methods, such as [1,2], satisfy our definition, with the only difference lying in the selection of distinct $\psi_i(\cdot)$ in Formula 3. Built upon the principles of graph prompt tuning, our method, GPF, is a general graph template that is theoretically applicable to any pre-training task. While the form of GPF may appear relatively simple, its effectiveness in bridging the gap between pre-training tasks and downstream tasks is theoretically guaranteed (Theorem 1). Traditional parameter tuning methods are primarily designed to optimize parameter efficiency. However, our method is designed with the primary goal of bridging the gap between pre-training tasks and downstream tasks. Therefore, our method should be categorized as a prompt tuning method that also offers advantages in terms of parameter efficiency.
>
> 2. (Weakness 2) The results for multi-class classification tasks.
>
> Not all experiments involved are conducted on binary classification tasks. The IMDB-M dataset illustrates a multi-class classification task on graphs, and you can refer to the corresponding results in Table 10 in the appendix. We have included additional results in B.2 of the global response to address your concerns about GPF's performance on multi-class tasks. These experimental results showcase the satisfactory performance of our methods in handling multi-class classification tasks, and we hope they can dispel your concern.
>
> In addition to comparing the results with fine-tuning, we have also conducted a comparison between GPF and existing graph prompting methods, as presented in Table 2. Furthermore, we have assessed GPF's performance against linear probing, as shown in Table 7, and compared GPF with other model tuning methods, as presented in Table 8. Through these extensive comparisons, we aim to evaluate GPF's performance comprehensively.
>
> 3. (Weakness 3) The modifications to the graph structure.
>
> Our method, GPF, operates on the feature space of the input graph, but it does not imply that we overlook modifications to the graph structure. While we focus on adding extra learnable vectors to the feature space, it is important to note that we have demonstrated the ability of this operation to achieve an equivalent effect to any structure modification. Detailed proofs regarding this equivalence can be found in Propositions 4 and 5 in the appendix. Consequently, our method does not need explicit structural modifications. There are several advantages to this simplification. The adjacency matrix, an N*N matrix, is much larger than the feature matrix. By working on the feature matrix, our method significantly reduces resource consumption and minimizes training difficulties.
>
> Furthermore, it is worth noting that our method, GPF, is not restricted to graph classification tasks alone. Section A.1 in the appendix provides detailed elaboration on how GPF extends to encompass node-wise tasks (node classification and link prediction). To address your concern regarding GPF's performance on node-wise tasks, we have included additional experimental results in B.1 of the global response.
>
> 4. (Weakness 4) Details about the experiment settings.
>
> The dimension of the basic prompt features matches that of the input graph node features. In our experiments, when applying the GCC framework, the dimension of the prompt features is set to 64, while for other frameworks, it is set to 300. You can find the relevant description in the main text at line 203 and Section B.4 in the appendix. Regarding the hyper-parameter $k$ of GPF-plus, it determines the number of prompt feature bases utilized. This parameter can be adjusted manually and vary across different datasets. You can find a detailed description of this parameter in Equation 9 and lines 300-301 of the main text. We are sorry to make you confused, and we have included an additional hyper-parameter list in the latest revision to provide more clarity on these parameters.
>
>
>
> We hope our explanation can dispel your concerns. If you have any other questions or concerns, please feel free to let us know.
>
>
>
> **Ref**
>
> [1] GPPT: Graph Pre-training and Prompt Tuning to Generalize Graph Neural Networks.
>
> [2] GraphPrompt: Unifying Pre-Training and Downstream Tasks for Graph Neural Networks.

---

> > ### Comment · Reviewer_GUF8 · 2023-08-18
> >
> > I'd like to express my gratitude to the authors for their thorough response. After careful consideration, I'm convinced that the authors have adeptly addressed all the concerns I raised. In light of this, I will revise my score to a 5. Thank you for your diligent efforts.

---

> > > ### Author Response · Authors · 2023-08-20
> > >
> > > Thanks for your support in our work. Your valuable feedback has made our work better.

---

### Official Review · Reviewer_aqDC · 2023-07-06

**Soundness:** 3 good
**Presentation:** 3 good
**Contribution:** 3 good
**Rating:** 7
**Confidence:** 4

**Summary:**

This paper proposes the Graph Prompt Feature (GPF) to improve Graph Neural Networks (GNNs) performance amidst scarce labeled data and low out-of-distribution generalization. GPF, a universal prompt-based tuning method, operates on the input graph's feature space and is applicable to any GNN architecture. The authors also present a stronger variant, GPF-plus, providing diverse prompted features for different nodes. Both methods show better performance than fine-tuning, validated through theoretical analyses and extensive experiments.


**Strengths:**

1. GPF and GPF-plus are universal, model-agnostic solutions that can be applied to any pre-trained GNN model, enhancing their general applicability in diverse scenarios.
2. The paper provides theoretical guarantees for the effectiveness of GPF and GPF-plus, strengthening the scientific rigor of the presented methods.
3. The authors demonstrate that their proposed methods outperform existing fine-tuning strategies and specialized prompt-based tuning methods in both full-shot and few-shot scenarios, showcasing their practical effectiveness.



**Weaknesses:**

1. The effectiveness of GPF and GPF-plus largely depends on the quality and representation of the feature space, which may not be optimally prepared in all real-world applications.
2. While the paper provides theoretical analysis for the methods' effectiveness, more detailed explanation or examples of the theoretical proofs could enhance the clarity and comprehensibility.
3. The real-world applicability and performance of the proposed methods could be further substantiated with experiments on a wider range of tasks and datasets.


**Questions:**

1. How do GPF and GPF-plus perform in scenarios where the feature space is noisy or inadequately represented? Are there strategies to mitigate these potential issues?
2. Could you provide more detailed explanation or real-world examples to further illustrate the theoretical guarantees of GPF and GPF-plus?
3. How would GPF and GPF-plus integrate with other advancements in GNNs or machine learning in general?

**Limitations:**

Not really.  Dear authors, please enumerate some limitations of your work.

---

> ### Author Rebuttal · Authors · 2023-08-09
>
> ### **Response**
>
> Dear reviewer aqDC,
> We appreciate your comments and your support for our work. We hope our response can address your concerns. Please find our detailed response below.
>
> 1. (Weakness 1 \& Question 1) Dealing with scenarios where the feature space is noisy or inadequately represented.
>
> Our proposed GPF and GPF-plus methods operate on the feature space of the input graph. We understand that your concern may lie in the fact that node features can be severely corrupted or contain unacceptable levels of noise in certain datasets. Notably, reducing dependence on node features often leads to better results in such cases. In these situations, we suggest replacing the raw features with synthetic features, such as positional encoding features [1,2] or Gaussian features [3], before applying GPF and GPF-plus to the generated node features. In Appendix B.7, we present experimental results on GCC that demonstrate the effectiveness of GPF on synthetic features. In this particular experiment, all node features are generated as positional encodings, and in this scenario, GPF still outperforms fine-tuning.
>
> 2. (Weakness 2 \& Question 2) Detailed explanation or examples of the theoretical proofs.
>
> Our theoretical analysis ensures the universal capability and effectiveness of our method. We appreciate your suggestion to include additional examples, and we have incorporated concrete examples in our latest revision to illustrate the theorems more comprehensively. Regarding Theorem 1, which establishes the universal capability of GPF, we have enhanced the theorem proofs by incorporating specific examples. These examples encompass existing graph templates like GPPT as well as intuitively designed graph templates. By demonstrating the equivalence of these templates to specific forms of GPF, we aim to provide readers with a deeper understanding of the universal capability of our method. As for Theorem 2, which showcases the effectiveness of GPF, we have manually designed a specific graph task to illustrate how GPF outperforms fine-tuning by providing a more optimal solution. This example serves as evidence that GPF can achieve superior performance in certain scenarios. Thanks for your detailed suggestion, and we believe that these enhancements and additional examples strengthen our theoretical claims.
>
> 3. (Weakness 3 \& Question 3) The integration with other advancements in GNN models.
>
> Your suggestion is indeed reasonable. We have included additional experimental results utilizing more powerful backbone models. You can find these results in B.1 of the global response.
>
>
>
> **Ref**
>
> [1] Equivariant and Stable Positional Encoding for More Powerful Graph Neural Networks.
>
> [2] On the Equivalence Between Positional Node Embeddings and Structural Graph Representations.
>
> [3] Random Features Strengthen Graph Neural Networks.

---

> ### Comment · Reviewer_aqDC · 2023-08-17
>
> I read the authors' response, and maintain the same rating of 7.

---

> > ### Author Response · Authors · 2023-08-20
> >
> > Thanks for your support in our work. Your valuable feedback has made our work better.

---

### Author Rebuttal · Authors · 2023-08-09

### **Global Response**

Dear all reviewers,
We appreciate your valuable comments on our work. We provide the following clarification and additional experimental results based on feedback.

**A. Contributions and influence**

We propose a universal graph prompt tuning method that can be applied to any pre-trained GNN models, and its effectiveness is theoretically guaranteed. By considering the impact of the pre-training task, our method is distinguished from conventional parameter tuning techniques. Importantly, our work is the pioneer in providing rigorous theoretical analysis on the effectiveness of graph prompting methods, offering valuable insights for future investigations in this field. We highlight an updated work, "All in One: Multi-task Prompting for Graph Neural Networks", which received the best research paper award of SIGKDD 2023 recently and successfully incorporated our innovative ideas with multi-task meta-learning, leading to remarkable performance improvements. Our work serves as its theoretical foundation, providing indispensable theoretical guarantees for their proposed method.

**B. Additional experiment results**

1. The performance of advanced backbone models on node classification.

We conduct experiments with advanced backbone models SUN(EGO+) [1] and 1-2-GNN [2] with subgraph aggregation on four node classification datasets pre-trained by Infomax. The following table presents the accuracy (%).

| SUN(EGO+) |      Cora      |    CiteSeer    |     PubMed     |   Ogbn-arxiv   |
| :-------: | :------------: | :------------: | :------------: | :------------: |
|    FT     |   83.15±0.23   |   71.10±0.27   |   79.11±0.21   |   70.89±0.11   |
|    GPF    |   83.46±0.33   |   72.81±0.67   |   79.91±0.42   |   70.92±0.09   |
| GPF-plus  | **83.70±0.45** | **72.89±0.53** | **80.79±0.95** | **70.95±0.17** |

| 1-2-GNN (Subgraph) |      Cora      |   CiteSeer    |     PubMed     |   Ogbn-arxiv   |
| :----------------: | :------------: | :-----------: | :------------: | :------------: |
|         FT         |   83.47±0.49   |  71.33±0.84   |   78.27±0.95   |   70.32±0.97   |
|        GPF         |   83.85±0.25   |  71.29±0.76   |   79.87±0.44   | **71.03±0.05** |
|      GPF-plus      | **84.01±0.87** | **71.56±0.10** | **79.95±0.20** |   70.82±0.40   |



2. The performance on multi-class classification.

We conduct experiments on the multi-class dataset Ogbg-ppa, and the following table presents the accuracy (%).

| Ogbg-ppa |    Infomax     |    EdgePred    |  AttrMasking   |  ContextPred   |      GCL       |
| :------: | :------------: | :------------: | :------------: | :------------: | :------------: |
|    FT    |   69.17±0.52   |   69.59±0.82   |   69.02±0.55   |   69.35±0.97   |   68.43±0.66   |
|   GPF    |   69.55±0.43   | **70.03±0.37** |   69.76±0.69   |   69.40±0.34   |   69.03±0.51   |
| GPF-plus | **69.95±0.27** |   69.73±0.37   | **69.88±0.43** | **69.51±0.37** | **69.11±0.63** |

Besides, some datasets in Table 1 and Table 2 of our paper contain multiple binary classification tasks. When we combine these binary classifications into a multi-class task and conduct experiments, we obtain the following ROC-AUC scores (%).

|          |          |     Tox21      |    ToxCast     |     SIDER      |    ClinTox     |
| :------: | :------: | :------------: | :------------: | :------------: | :------------: |
| Infomax  |    FT    |   63.84±0.80    |   52.18±0.19   |   52.55±0.51   |   60.11±0.95   |
|          |   GPF    |   64.17±0.95   |   53.35±0.47   | **55.21±0.19** |   63.81±0.33   |
|          | GPF-plus | **64.28±0.13** | **53.54±0.28** |   54.81±0.54   | **65.38±0.94** |
| EdgePred |    FT    |   63.75±0.72   |   52.36±0.09   |   53.45±0.75   |   59.30±0.44   |
|          |   GPF    |   64.85±0.18   |   52.78±0.55   |   56.35±0.05   | **59.60±0.51** |
|          | GPF-plus | **65.13±0.47** | **52.95±0.99** | **56.61±0.30** |   58.82±0.45   |



3. Comparison with the adapter.

We conduct experiments with an adapter [3] before the final linear head. The final output representation dimension of the adapter is the same as the output representation dimension of the pre-trained model. The following table presents the ROC-AUC scores (%).

|          |          |      BBBP      |     Tox21      |    ToxCast     |     SIDER      |    ClinTox     |      BACE      |
| :------: | :------: | :------------: | :------------: | :------------: | :------------: | :------------: | :------------: |
| Infomax  | Adapter  |   64.19±0.46   |   76.30±0.02    |    63.21±0.20    |   62.14±0.39   |   69.33±0.01   |   80.51±0.26   |
|          |   GPF    |   66.83±0.86   |   79.09±0.25   |   66.10±0.53    | **66.17±0.81** |   73.56±3.94   |   83.60±1.00   |
|          | GPF-plus | **67.17±0.36** | **79.13±0.70** | **66.35±0.37** |   65.62±0.74   | **75.12±2.45** | **83.67±1.08** |
| EdgePred | Adapter  |   63.95±0.25   |   75.21±0.41   |   63.72±0.26   |   61.78±0.23   |   67.12±0.28   |   79.45±0.40    |
|          |   GPF    | **69.57±0.21** |   79.74±0.03   |   65.65±0.30    |   67.20±0.99    | **69.49±5.17** |   81.57±1.08   |
|          | GPF-plus |   69.06±0.68   | **80.04±0.06** | **65.94±0.31** | **67.51±0.59** |   68.80±2.58   | **81.75±2.09** |



Due to limited time, some experiments are not validated under all pre-training strategies and datasets. We will complete them later.



**Ref**

[1] Understanding and extending subgraph gnns by rethinking their symmetries.

[2] Weisfeiler and leman go neural: Higher-order graph neural networks.

[3] Parameter-Efficient Transfer Learning for NLP.

---

### Decision · Program_Chairs · 2023-09-21

**Decision:**

Accept (poster)

**Comment:**

The paper explores the adaptation of pre-trained graph neural networks for downstream tasks. It introduces a simple method referred to as 'prompt tuning.' This approach involves enhancing the graph node features with a learnable component. The paper later presents a refinement of this concept. The method is termed 'universal' because it can be applied to GNNs pre-trained with various objectives. The authors provide both theoretical guarantees and experimental evaluations.

The reviewers raised several concerns about the presentation, including a lack of intuition regarding the reasons for improvements over the baselines, the relationship with prompt tuning in Language Model Models, limited experiments. The authors responded by providing comprehensive explanations and conducting additional experiments that persuaded the reviewers to increase their scores and lean toward accepting the paper. Overall, the proposed method offers a general method with good performance that covers a variety of GNNs adaptation setups. I recommend an accept.